# Occurrence and Morpho-Molecular Identification of *Botryosphaeriales* Species from Guizhou Province, China

**DOI:** 10.3390/jof7110893

**Published:** 2021-10-22

**Authors:** Asha J. Dissanayake, Ya-Ya Chen, Ratchadawan Cheewangkoon, Jian-Kui Liu

**Affiliations:** 1School of Life Science and Technology, Center for Informational Biology, University of Electronic Science and Technology of China, Chengdu 611731, China; asha.janadaree@yahoo.com; 2Institute of Crop Germplasm Resources, Guizhou Academy of Agricultural Sciences, Guiyang 550006, China; wmlove@163.com; 3Guizhou Key Laboratory of Agricultural Biotechnology, Guizhou Academy of Agricultural Sciences, Guiyang 550006, China; 4Department of Entomology and Plant Pathology, Faculty of Agriculture, Chiang Mai University, Chiang Mai 50200, Thailand; ratchadawan.c@cmu.ac.th

**Keywords:** asexual morph, phylogeny, saprobes, sexual morph, three new species, woody hosts

## Abstract

*Botryosphaeriales* is an important order of diverse fungal pathogens, saprobes, and endophytes distributed worldwide. Recent studies of *Botryosphaeriales* in China have discovered a broad range of species, some of which have not been formerly described. In this study, 60 saprobic isolates were obtained from decaying woody hosts in southwestern China. The isolates were compared with other species using morphological characteristics, and available DNA sequence data was used to infer phylogenetic analyses based on the internal transcribed spacer (ITS), large subunit rRNA gene (LSU), and translation elongation factor 1-α (*tef*) loci. Three novel species were illustrated and described as *Botryobambusa* *guizhouensis*, *Sardiniella* *elliptica*, and *Sphaeropsis* *guizhouensis*, which belong to rarely identified genera within *Botryosphaeriaceae*. *Botryobambusa* *guizhouensis* is the second species identified from the respective monotypic genus. The previously known species were identified as *Aplosporella hesperidica*, *Barriopsis tectonae*, *Botryosphaeria dothidea, Diplodia mutila*, *Di. neojuniperi*, *Di. pseudoseriata*, *Di*. *sapinea*, *Di*. *seriata*, *Dothiorella sarmentorum*, *Do*. *yunnana*, *Lasiodiplodia pseudotheobromae*, *Neofusicoccum parvum*, *Sardiniella celtidis*, *Sa.* *guizhouensis*, and *Sphaeropsis citrigena*. The results of this study indicate that numerous species of *Botryosphaeriales* are yet to be revealed in southwestern China.

## 1. Introduction

The *Botryosphaeriales* are among the most widespread, common, and important fungal pathogens of woody plants. Many are known to exist as endophytes in healthy plant tissues and also as saprobes in dead tree materials. This fungal order has gone through significant revisions, and several new families, genera, and species have been introduced or synonymized over the last decade, mainly on the basis of combined morphological and multiple gene sequence data [1,2,3,4,5,6,7,8,9]. Schoch et al. [10] introduced the order *Botryosphaeriales* to accommodate the single family *Botryosphaeriaceae*. In the “Outline of Ascomycetes” [7], nine families (*Aplosporellaceae*, *Botryosphaeriaceae*, *Melanopsaceae*, *Phyllostictaceae*, *Planistromellaceae*, *Saccharataceae*, *Septorioideaceae*, *Endomelanconiopsidaceae*, and *Pseudofusicoccaceae*) were recognized in *Botryosphaeriales*. Phillips et al. [3] revised the order and accepted *Aplosporellaceae*, *Botryosphaeriaceae*, *Melanopsaceae*, *Phyllostictaceae*, *Planistromellaceae*, and *Saccharataceae*, while *Endomelanconiopsidaceae*, *Pseudofusicoccaceae*, and *Septorioideaceae* were considered as synonyms of *Botryosphaeriaceae*, *Phyllostictaceae*, and *Saccharataceae*, respectively. We followed this last taxonomical revision in our study.

Presently, the order *Botryosphaeriales* comprises 33 genera [3,9]. *Alanomyces* and *Aplosporella* are the only two genera accepted within the family *Aplosporellaceae*. The family *Botryosphaeriaceae* currently comprises 22 genera: *Alanphillipsia*, *Barriopsis*, *Botryobambusa*, *Botryosphaeria*, *Cophinforma*, *Diplodia*, *Dothiorella*, *Endomelanconiopsis*, *Eutiarosporella*, *Lasiodiplodia*, *Macrophomina*, *Marasasiomyces*, *Mucoharknessia*, *Neodeightonia*, *Neofusicoccum*, *Neoscytalidium*, *Oblongocollomyces*, *Phaeobotryon*, *Sakireeta*, *Sardiniella*, *Sphaeropsis*, and *Tiarosporella* [3,9]. *Melanopsaceae* accommodates only one genus, *Melanops*, which was supported by several phylogenetic analyses [3,4,11], while *Phyllostictaceae* includes two genera, *Phyllosticta* and *Pseudofusicoccum* [3,4,11]. *Planistromellaceae* was revised to accommodate *Kellermania* and *Umthunziomyces* [3]. *Saccharataceae* comprises three genera, *Pileospora*, *Saccharata*, and *Septorioides* [3,4].

*Botryosphaeriales* species cause blight, canker, dieback, and fruit rots on a variety of woody perennials globally [5,12]. In China, infections related to *Botryosphaeriales* have been described on numerous hosts such as grapes [13,14,15], *Caragana arborescens* [16], *Cercis chinensis* [17], *Eucalyptus* [12], Chinese hackberry [18], blueberry [19,20], forest trees [21,22], and various other woody hosts. Hence, the aim of this study was to characterize the *Botryosphaeriales* taxa associated with woody hosts in southwestern China based on morphology, DNA sequence data, and phylogeny.

## 2. Materials and Methods

### 2.1. Collection of Specimens, Isolation, Morphology, and Culture Characteristics

From 2017 to 2019, specimens were collected in field investigations of numerous decomposing woody hosts in Fanjing mountain (Tongren District), Forest Park (Chishui District), Huaxi wetland park, Xiaochehe wetland park (Guiyang District), Maolan natural reserve (Libo District), Suiyang broad water nature reserve, and Xingyi Wanfenglin in the Karst region of Guizhou province (Table 1). Samples were placed into ziplock plastic bags, relevant data (location, date, etc.) were documented, and samples were taken into the laboratory.

Morphological observations of conidiomata or ascostromata were carried out using a Motic SMZ 168 series stereomicroscope and photographed using a Nikon E80i microscope-camera system. Tarosoft^®^ Image Framework was used to measure morphological characters as in Liu et al. [23], and images included in figures were processed with Adobe Photoshop cs v. 5. To isolate single spores, the procedure according to Chomnunti et al. [24] was followed. Spores germinated on water agar (WA) for 12–24 h were examined and then transferred to potato dextrose agar (PDA) media (OXOID CM0139). Obtained pure cultures were incubated at 25 °C for two weeks, and colony characteristics and morphology of fungal structures were examined for a total of 60 isolates. According to Rayner [25], colony color was inspected after 5–10 days of progression on PDA at 25 °C. Approximately 20 ascomata/conidiomata, 25 asci, and 50 conidia/ascospores were measured to obtain the mean size/length. Shape, color, and presence or absence of the mucous sheath of conidia/ascospores were also documented.

Herbarium specimens were deposited at the HKAS (Herbarium of Cryptogams, Kunming Institute of Botany Academia Sinica Kunming, China) and GZAAS (Herbarium of Guizhou Academy of Agricultural Sciences, Guiyang, China), while living cultures were deposited in the CGMCC (China General Microbiological Culture Collection Center in Beijing, China) and GZCC (Guizhou Culture Collection in Guiyang, China) (Table 1).

### 2.2. DNA Extraction and Molecular Based Amplification

About 10 mg of aerial mycelia were scraped from 5 day-old isolates grown on PDA medium at 25 °C for the extraction of total genomic DNA using an Extraction Kit of Biospin Fungus Genomic DNA (BioFlux^®^, Hangzhou, China) according to the manufacturer’s protocol (Hangzhou, China). For initial species confirmation, the internal transcribed spacer (ITS) region was sequenced for all isolates. The BLAST tool (https://blast.ncbi.nlm.nih.gov/Blast.cgi, accessed on 14 August 2020) was used to compare the resulting sequences with those in GenBank. After confirmation of *Botryosphaeriales* species, two additional gene regions coding for translation elongation factor 1-α (*tef*) and large subunit rRNA gene (LSU) were sequenced as in Dissanayake et al. [5]. The primer pairs and amplification conditions for each of the above-mentioned gene regions are provided in Table 2. A Bio-Rad C1000 thermal cycler was used to conduct the PCR reactions. The resulting PCR products were visualized on a 1% agarose gel stained with ethidium bromide under UV light by a Gel Doc^TM^ XR Molecular Imager (Bio-Rad, USA). All positive amplicons were sequenced by Shanghai Sangon Biological Engineering Technology and Services Co., Ltd. (Shanghai, China).

### 2.3. Sequence Alignment and Phylogenetic Analyses

Sequence quality was assured by inspecting the chromatograms using BioEdit v. 5 [29]. Sequences were obtained with both forward and reverse primers, and consensus sequences were obtained using DNAStar v. 5.1 (DNASTAR, Inc.). The sequence data generated in this study have been deposited in GenBank (Table 1).

Reference sequences of ITS, *tef*, and LSU were retrieved from NCBI GenBank, referring to recent publications [3,4,5,6,9] (Table 3) to conduct phylogenetic analyses. The reference sequences were aligned with the sequences obtained in this study (Table 1) using MAFFT (http://www.ebi.ac.uk/Tools/msa/mafft/, accessed on 22 December 2020) [30], then manually adjusted, and phylogenetic relationships were inferred with maximum likelihood (ML), maximum parsimony (MP), and Bayesian inference (BI) using procedures provided in detail by Dissanayake et al. [31]. An overview phylogenetic tree for the order *Botryosphaeriales* was constructed using ITS, LSU, and *tef* sequence data as some families in *Botryosphaeriales* (except *Botryosphaeriaceae*) comprise only ITS and LSU sequences. Separate phylogenetic trees of the diverse genera (*Botryosphaeria*, *Diplodia*, *Dothiorella*, *Lasiodiplodia*, and *Neofusicoccum*) were constructed by combining ITS and *tef* sequences.

Alignments generated in this study were submitted to TreeBASE (https://treebase.org/treebase-web/home.html, accessed on 18 August 2021). The submission numbers and reviewer access URL for each alignment are provided in Table 4. Taxonomic novelties were submitted to the Faces of Fungi database [32] and Index fungorum (http://www.indexfungorum.org, accessed on 5 August 2021). New species were established based on the guidelines provided by Jeewon and Hyde [33].

## 3. Results

### 3.1. Phylogenetic Analyses

Sixty isolates obtained from various decaying woody hosts in various locations in Guizhou province were primarily recognized by colony characteristics, such as abundant greenish black aerial mycelia on PDA medium. The ITS gene region sequences compared with those in GenBank using the BLAST tool exhibited 95–99% similarity to known *Botryosphaeriales* species, and these closely related known species were included in the phylogenetic analysis. All details of the alignments (ITS, LSU, *tef* alignment of the overview phylogenetic tree for the order *Botryosphaeriales* and ITS and *tef* alignments for the genera *Botryosphaeria*, *Diplodia*, *Dothiorella*, *Lasiodiplodia*, and *Neofusicoccum*) are provided in Table 4. The best-scoring RAxML tree (Figure 1) is presented as the MP and BI methods produced trees with topologies similar to those of ML.

Six different phylogenetic trees were constructed for the 60 isolates obtained in this study. Twelve isolates (20% of total isolates) were treated together in an overview phylogenetic tree and seven of them did not cluster with any known *Botryosphaeriales* species, thus, three novel species were identified based on the morphological and phylogenetic evidence (Figure 1). In this phylogeny, the isolates obtained in the study were clustered with *Aplosporella hesperidica* (Figure 2), *Barriopsis tectonae* (Figure 3), *Botryobambusa guizhouensis* sp. nov. (Figure 4), *Sardiniella celtidis* (Figure 5), *Sardiniella elliptica* sp. nov. (Figure 6 and Figure 7), *Sardiniella guizhouensis* (Figure 8), *Sphaeropsis citrigena*, and *Sphaeropsis guizhouensis* sp. nov (Figure 9).

Twenty-three isolates (38.3% of total isolates) belong to the genus *Botryosphaeria*, and all of them clustered with *B. dothidea* (Figure 10). Six isolates (10% of total isolates) belong to the genus *Diplodia* and were identified as *Di. mutila*, *Di. neojuniperi*, *Di. pseudoseriata*, *Di. sapinea*, and *Di. seriata* (Figure 11). Two isolates (3.3% of total isolates) were identified as species of *Dothiorella* (*Do. sarmentorum* and *Do. yunnana*, Figure 12), while five isolates (8.3% of total isolates) belong to the genus *Lasiodiplodia* (*L. pseudotheobromae*, Figure 13). All twelve isolates (20% of total isolates) of the genus *Neofusicoccum* were identified as *N. parvum* (Figure 14).

### 3.2. Taxonomy

***Aplosporella hesperidica*** Speg., Anal. Soc. cient. argent. 13: 18 (1882) (Figure 2).

Index Fungorum number: IF218239; Facesoffungi number: FoF07830.

*Saprobic* on decaying wood. **Sexual morph:** Not observed. **Asexual morph:** *Conidiomata* 220–360 × 420–610 µm ($\bar{x}$ = 320 × 550 µm, n = 20), solitary, dark brown, immersed to semi-immersed, erumpent, multiloculate, locules separated by pale brown *textura prismatica*. *Ostiole* 60–80 µm diam., central. *Peridium* 75–150 µm (6–10 cell-layers), outer layers composed of dark brown *textura angularis*, becoming hyaline towards the inner region. *Conidiophores* reduced to conidiogenous cells. *Conidiogenous cells* 6–11 × 2.5–3 µm ($\bar{x}$ = 8 × 2.5 µm, n = 20), holoblastic, hyaline, cylindrical to doliiform, smooth-walled, proliferating percurrently with 1–3 annellations near the apex. *Paraphyses* 35–95 × 4–8 µm ($\bar{x}$ = 60 × 5 µm, n = 20), wide at the base, 1–3 µm wide in the upper part, hyaline, smooth-walled, septate, branched below. *Conidia* 17–25 × 10–18 µm ($\bar{x}$ = 20 × 12 µm, n = 50), aseptate, initially hyaline, smooth-walled, broadly ellipsoidal to subcylindrical, with rounded ends, becoming dark brown (black in mass), prominently verruculose before discharge from pycnidia.

**Culture characteristics**: Conidia germinate on WA within 12 h at room temperature. Colonies on PDA after five days at 25 °C become olivaceous to grey-olivaceous in the center, olivaceous-buff to greenish-olivaceous towards the margin. Aerial mycelium appressed, floccose, white to smoke grey. Colonies flat with undulate edge, 38 mm diameter after two days, reaching the edge of the Petri dish within 10 days.

**Material examined: China**, Guizhou province, Tongren District, Fanjing mountain, on decaying woody host, July 2018, Y. Y. Chen, GZAAS 19-1814, living culture GZCC 19-0095.

**Notes:** Our sample morphologically lines up with the description of *Aplosporella hesperidica* provided by Spegazzini [34] as it has erumpent, black conidiomata and brown, smooth-walled, oblong conidia with overlapping spore dimensions of 22–25 × 9–11 µm. It is identical to *A. hesperidica* based on morphology and phylogeny (Figure 1). This is the first time *A. hesperidica* has been reported in China.

**Figure 2 jof-07-00893-f002:**
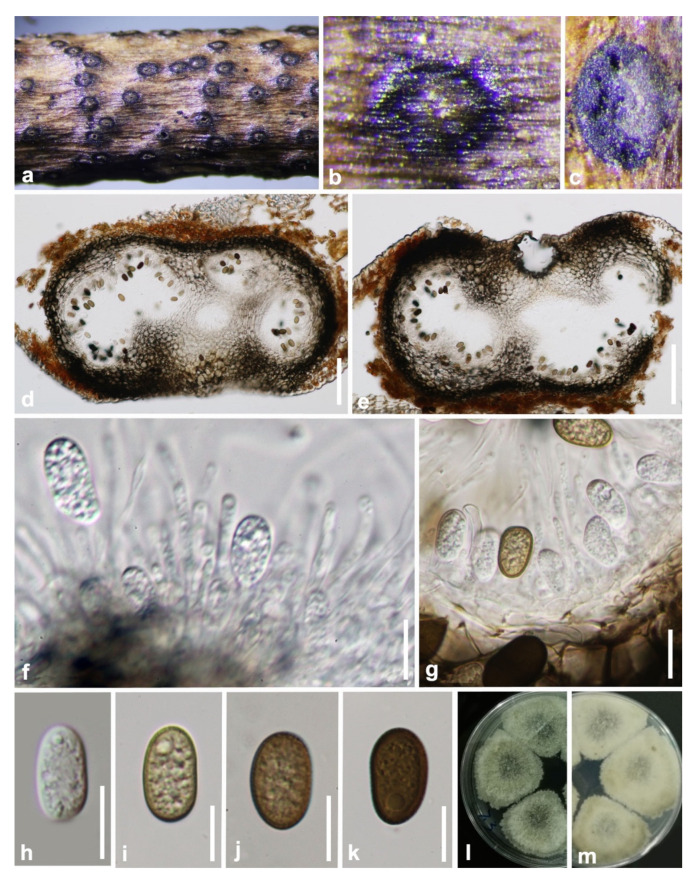
***Aplosporella hesperidica*** (GZAAS 19-1814). (**a**–**c**) Conidiomata on host surface. (**d**,**e**) Vertical hand sections of multiloculate conidiomata. (**f**,**g**) Developing conidia attached to conidiogenous cells. (**h**) Hyaline immature conidium. (**i**–**k**) Mature brown conidia. (**l**,**m**) Five-day-old culture on PDA (OXOID CM0139) from above and below. Scale bars: (**d**,**e**) = 100 μm, (**f**–**k**) = 10 μm.

***Barriopsis tectonae*** Doilom, L.A. Shuttlew., and K.D. Hyde, Phytotaxa 176: 84 (2014) (Figure 3).

Index Fungorum number: IF808202; Facesoffungi number: FoF09644.

*Saprobic* on decaying wood. **Sexual morph:** *Ascostromata* 286–420 × 258–349 μm ($\bar{x}$ = 350 × 318 µm, n = 20), black, immersed, aggregated or clustered, scattered, multi-loculate, composed of one or up to three globose ascomata in each ascostroma, erumpent through the bark at maturity, discoid to pulvinate or hemispherical, discrete or wide-spreading. *Peridium* composed of several layers of dark brown-walled cells of *textura angularis*. *Pseudoparaphyses* 2.5–3.5 μm wide, hyphae-like, septate, embedded in a gelatinous matrix. *Asci* 107–154 × 34–41 ($\bar{x}$ = 129 × 36 µm, n = 25), eight-spored, bitunicate, fissitunicate, clavate to sub-clavate, broad, with a short pedicel and apically rounded with an ocular chamber. *Ascospores* 31–34 ×14–15 µm ($\bar{x}$ = 32 × 15 µm, n = 50), biseriate, brown to dark brown, aseptate, ellipsoid-oval, inequilateral, slightly curved, widest in the median, ends rounded, light brown in the center, smooth or verrucose, without a gelatinous sheath. **Asexual morph:** Not observed.

**Culture characteristics:** Ascospores germinate on WA within 18 h. Colonies growing on PDA reach 2 cm diameter after five days at 25 °C. Effuse, velvety with entire to slightly undulate edge. Blackish green to black.

**Material examined: China**, Guizhou province, Libo District, Maolan natural reserve, on decaying woody host, July 2017, Y. Y. Chen, GZAAS 19-1985, living culture GZCC 19-0266.

**Notes:** In the phylogenetic analysis, an isolate obtained in this study (GZCC 19-0266) was grouped with the ex-type isolate of *Barriopsis tectonae* (Figure 1) with absolute bootstrap support (ML/MP/BI = 100/100/1.0). This isolate is morphologically similar to *Ba. tectonae* as of the report by Doilom et al. [35] with erumpent, black ascostromata and overlapping biseriate, brown, aseptate, ellipsoid, thick-walled conidia with overlapping spore dimensions of 29–33 × 13–15 μm ($\bar{x}$ = 30 × 14 μm, n = 30). It is identical to *Ba. tectonae* based on morphology and phylogeny (Figure 1). We therefore identify our isolate as *Ba. tectonae* based on phylogenetic analyses, and the isolate is introduced here as a new locality record from Guizhou province, China. This is the first time *Ba. tectonae* has been reported in China.

**Figure 3 jof-07-00893-f003:**
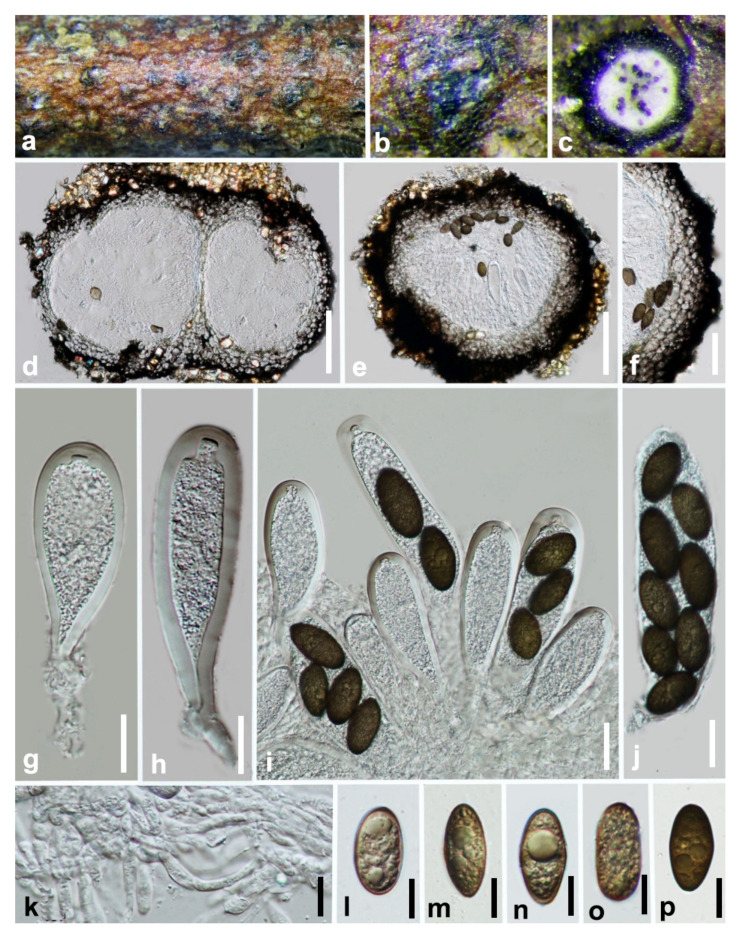
***Barriopsis tectonae*** (GZAAS 19-1985). (**a**,**b**) Appearance of ascomata on wood. (**c**) Horizontal section of the ascomata. (**d**,**e**) Vertical section of ascomata. (**f**) Peridium. (**g**,**h**) Immature asci. (**i**) Immature and mature asci. (**j**) Mature asci. (**k**) Pseudoparaphyses. (**l**–**p**). Brown ascospores. Scale bars: (**d**) = 50 μm, (**e**) = 100 μm, (**f**) = 50 μm, (**g**–**j**) = 20 μm, (**k**–**p**) = 10 μm.

***Botryobambusa guizhouensis*** Y.Y. Chen, A. J. Dissanayake, and Jian K. Liu., ***sp*. *nov*** (Figure 4).

Index Fungorum number: IF558473; Facesoffungi number: FoF09645.

**Etymology:** Name refers to the location where the fungus was collected, Guizhou, China.

**Holotype:** HKAS 112600.

*Saprobic* on a decaying bamboo. **Sexual morph:** *Ascostromata* 218–340 × 210–420 μm ($\bar{x}$ = 275 × 345 μm, n = 20), black, immersed or erumpent, gregarious, uniloculate, locules globose to subglobose, coriaceous. *Peridium* 36–60 μm, comprises several layers of cells *textura angularis*, broader at the base, outer layers dark to dark brown and thick-walled, inner layers hyaline and thin-walled. *Asci* 78–115 × 12–16 μm ($\bar{x}$ = 94.5 × 14.5 μm, n = 25), eight-spored, bitunicate, fissitunicate, clavate to cylindro-clavate, usually wider at the apex, pedicellate, apically rounded with well-developed ocular chamber. *Ascospores* 13–22 × 8–11 μm ($\bar{x}$ = 17.5 × 9 μm, n = 50), uniseriate at the base or irregularly biseriate, hyaline, aseptate, ellipsoidal to obovoid, thick-walled, surrounded by distinctive structured mucilaginous sheath. **Asexual morph:** Not observed.

**Culture characteristics:** Ascospores germinate on WA within 24 h. Colonies growing on PDA reach a 5 cm diameter after five days at 25 °C. Fast growing; white in the first few days, become grey to green-black after one week. Reverse grey to black, flattened, fairly dense, surface smooth with crenate edge.

**Material examined: China**, Guizhou province, Chishui District, Forest Park, on decaying bamboo, July 2019, Y. Y. Chen, 171, (HKAS 112600, holotype); ex-type living culture CGMCC 3.20348; *ibid.*, (GZAAS 20-0718, paratype), living culture GZCC 19-0734.

**Notes:***Botryobambusa guizhouensis* formed a distinct clade with absolute support (ML/MP/BI = 100/100/1.0) and differed from its closely related species *Bo. fusicoccum* in the concatenated alignment by 7/680 bp in ITS, 4/803 bp in LSU, and 5/479 bp in *tef*. Morphologically, *Bo. guizhouensis* differs from *Bo. fusicoccum* in having longer asci (94.5 × 14.5 μm vs. 60 × 15.5 μm) and larger ascospores (17.5 × 9 μm vs. 11.5 × 6 μm).

**Figure 4 jof-07-00893-f004:**
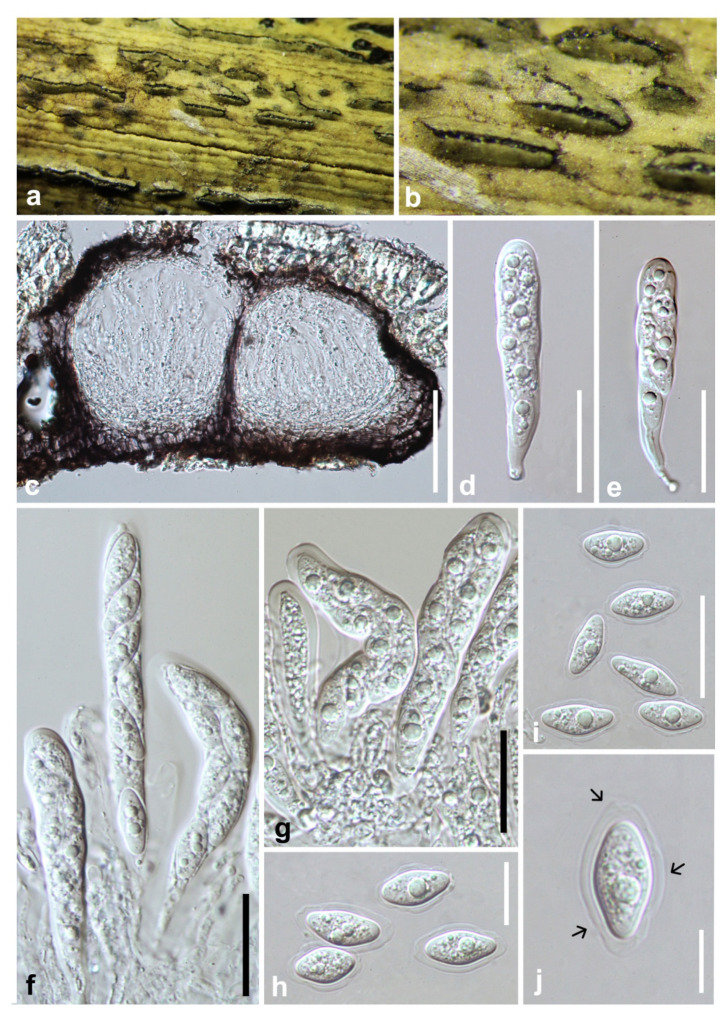
***Botryobambusa guizhouensis*** (HKAS 112600, holotype). (**a**,**b**) Appearance of ascomata on bamboo. (**c**) Vertical section of ascomata. (**d**,**e**) Immature asci. (**f**,**g**) Immature and mature asci. (**h**–**j**) Hyaline, aseptate ascospores enclosed with mucilaginous sheath. Scale bars: (**c**) = 50 μm, (**d**–**g**) = 100 μm, (**h**,**i**) = 20 μm, (**j**) = 10 μm.

***Sardiniella celtidis*** Dissan., Camporesi and K.D. Hyde, Fungal Divers 87: 12 (2017) (Figure 5).

Index Fungorum number: IF552896; Facesoffungi number: FoF02732.

*Saprobic* on a decaying host. **Sexual morph:** *Ascostromata* 210–300 × 275–340 μm ($\bar{x}$ = 270 × 310 μm, n = 20), dark brown to black, globose, immersed in the substrate, partially erumpent at maturity, ostiolate. *Ostiole* circular, central. *Peridium* 48–72 μm thick, composed of dark brown thick-walled cells of *textura angularis*, becoming thin-walled and hyaline towards the inner region. *Pseudoparaphyses* 3–5 μm wide, thin-walled, hyaline. *Asci* 62–90 × 26–34 μm ($\bar{x}$ = 78 × 30 μm, n = 20), four- to eight-spored, bitunicate, cylindric-clavate, endotunica thick-walled, with a well-developed ocular chamber. *Ascospores* 19–27 × 15–18 μm ($\bar{x}$ = 23 × 16 μm, n = 50), 1-septate, irregularly biseriate, dark brown, oblong to ovate, widest in center, straight, moderately thick-walled, surface smooth. **As****exual morph:** Not observed.

**Culture characters**: Ascospores germinate on WA within 24 h. Colonies on PDA reach a 2 cm diameter after five days at 25 °C. Mycelium velvety and moderately fluffy with an irregular margin, surface initially white and later turning dark olivaceous from the middle of the colony and dark grey in reverse.

**Material examined**: **China**, Guizhou province, Guiyang city, Xingyi Wanfenglin, on decaying woody host, June 2019, Y.Y. Chen, GZAAS 19-1967, living culture GZCC 19-0248.

**Notes:** Our sample is phylogenetically identical to *Sardiniella celtidis* (Figure 1). Only the asexual morph of *Sa. celtidis* was provided when Hyde et al. [36] introduced this species. Here, we provide the sexual morph of *Sa*. *celtidis*.

**Figure 5 jof-07-00893-f005:**
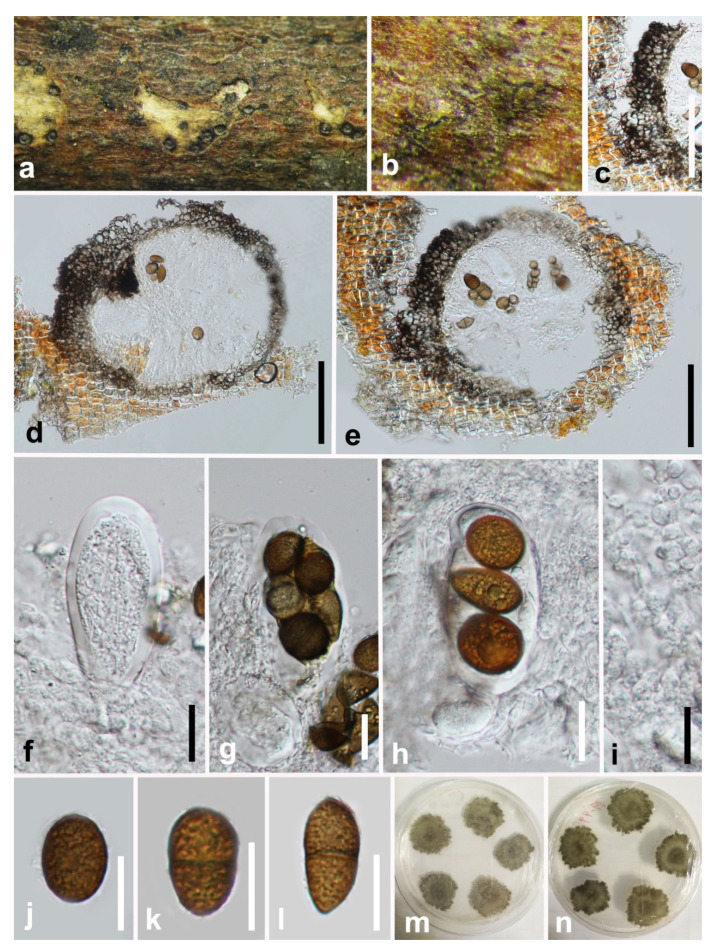
***Sardiniella celtidis*** (GZAAS 19-1967). (**a**,**b**) Appearance of ascomata on decaying wood. (**c**) Peridium. (**d**,**e**) Vertical section of ascomata. (**f**) Immature ascus. (**g**,**h**) Mature asci. (**i**) Pseudoparaphyses. (**j**–**l**) Brown, one-septate ascospores. (**m**,**n**) Five-day-old culture on PDA (OXOID CM0139) from above and below. Scale bars: (**c**–**e**) = 100 μm, (**f**–**l**) = 20 μm.

***Sardiniella elliptica*** Y.Y. Chen, A. J. Dissanayake, and Jian K. Liu., ***sp*. *nov*** (Figure 6 and Figure 7).

Index Fungorum number: IF558474; Facesoffungi number: FoF09646.

**Etymology:** Named referring to the shape of the conidiospores.

**Holotype:** HKAS 112594.

*Saprobic* on decaying host. **Sexual morph:** *Ascostromata* 280–390 × 295–340 μm ($\bar{x}$ = 340 × 310 μm, n = 20), dark brown to black, globose, submerged in the substrate, partially erumpent at maturity, ostiolate. *Ostiole* circular, central. *Peridium* 30–48 μm thick, composed of dark brown thick-walled cells of *textura angularis*, becoming thin-walled and hyaline towards the inner region. *Pseudoparaphyses* 2–3 μm wide, thin-walled, hyaline. *Asci* 71–93 × 19–24 μm, ($\bar{x}$ = 86 × 22 μm, n = 25), 4(–8)-spored, bitunicate, cylindric-clavate, endotunica thick-walled, with a well-developed ocular chamber. *Ascospores* 26–33 × 9–12 μm ($\bar{x}$ = 29 × 11 μm, n = 50), irregularly biseriate, initially hyaline and becoming dark brown, oblong to ovate, widest in center, straight, moderately thick-walled, surface smooth. **As****exual morph:** Appearing as subepidermal black spots on the substrate with black margins, with circular sunken perforation through the bark. *Conidiomata* 190–240 × 274–310 µm ($\bar{x}$ = 220 × 290 µm, n = 20), pycnidial, immersed, arranged singly or in small groups within the bark, globose to subglobose, dark brown to black, solitary or gregarious. *Ostiole* central. *Peridium* 27–35 µm thick, comprising dark brown to hyaline, multi-layered (3–5 layered), thick-walled cells of *textura angularis*. *Conidiogenous cells* lining the inner surface of the conidioma, hyaline, short obpyriform to subcylindrical. *Conidia* 25–32 × 10–13 µm ($\bar{x}$ = 28 × 12 µm, n = 50), ellipsoid to obovoid, immature conidia hyaline, mature conidia becoming medium to dark brown.

**Culture characters**: Colonies on PDA reaching a 70 mm diameter after five days at 25 °C. Mycelium velvety and moderately fluffy with an irregular margin. Surface initially white and later turning dark olivaceous from the middle of the colony and dark grey in reverse.

**Material examined**: **China**, Guizhou province, Guiyang District, Huaxi wetland park, on decaying woody host, April 2017, Y.Y. Chen, 18-76 (HKAS 112594, holotype); ex-type living culture CGMCC 3.20349; *ibid*., Libo District, Maolan natural reserve, July 2017, 19-120 (GZAAS 19-1855, paratype), living culture GZCC 19-0262; *ibid*., Xingyi Wanfenglin, June 2019, 19-96 (GZAAS 19-1838, paratype), living culture GZCC 19-0245.

**Notes:** Three isolates of *Sardiniella elliptica* clustered together with *Sa. celtidis, Sa. guizhouensis*, and *Sa. urbana* and formed a well-supported clade representing the genus *Sardiniella*; thus, it can be recognized as a distinct lineage within *Sardiniella*. *Sardiniella elliptica* can be distinguished from the above closely related species based on ITS and *tef* loci for *Sa. celtidis* (5/680 bp in ITS, 21/479 bp in *tef*) and *Sa. urbana* (5/680 bp in ITS, 8/803 bp in LSU, 27/479 bp in *tef*). In addition, *Sa. elliptica* can be morphologically distinguished from other known *Sardiniella* species based on its aseptate mature ascospores.

**Figure 6 jof-07-00893-f006:**
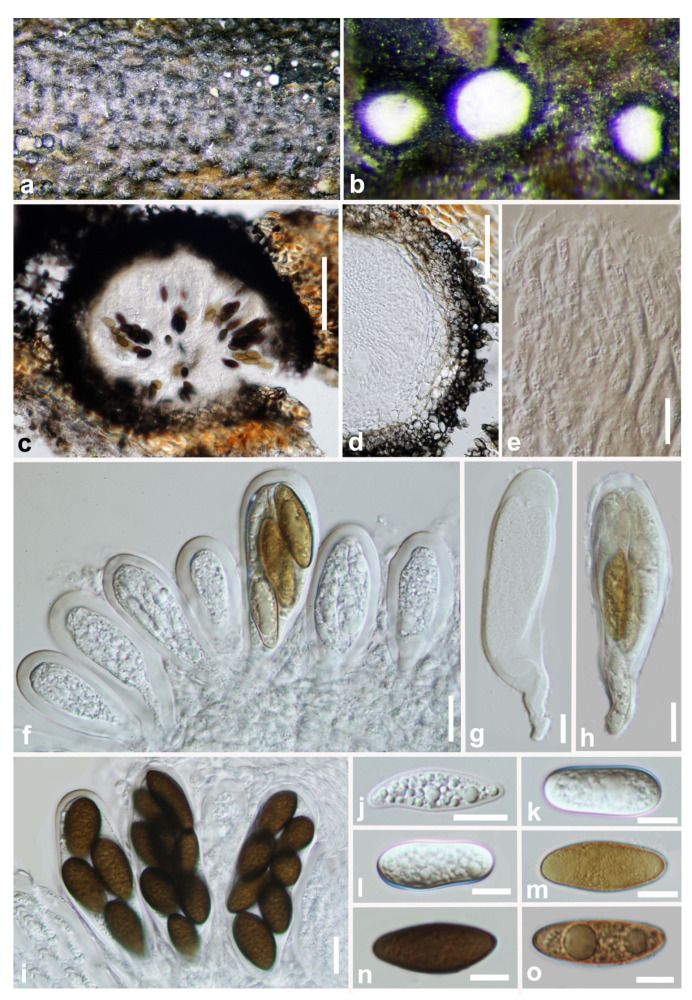
***Sardiniella elliptica*** (HKAS 112594, holotype, sexual morph). (**a**,**b**) Appearance of ascomata on decaying wood. (**c**) Vertical section of ascomata. (**d**) Peridium. (**e**) Pseudoparaphyses. (**f**) Immature and mature asci. (**g**) Immature ascus. (**h**,**i**) Mature asci. (**j**–**l**) Immature hyaline ascospores. (**m**–**o**) Mature, brown, aseptate ascospores. Scale bars: (**c**) = 100μm, (**d**) = 50 μm, (**e**–**o**) = 10 μm.

**Figure 7 jof-07-00893-f007:**
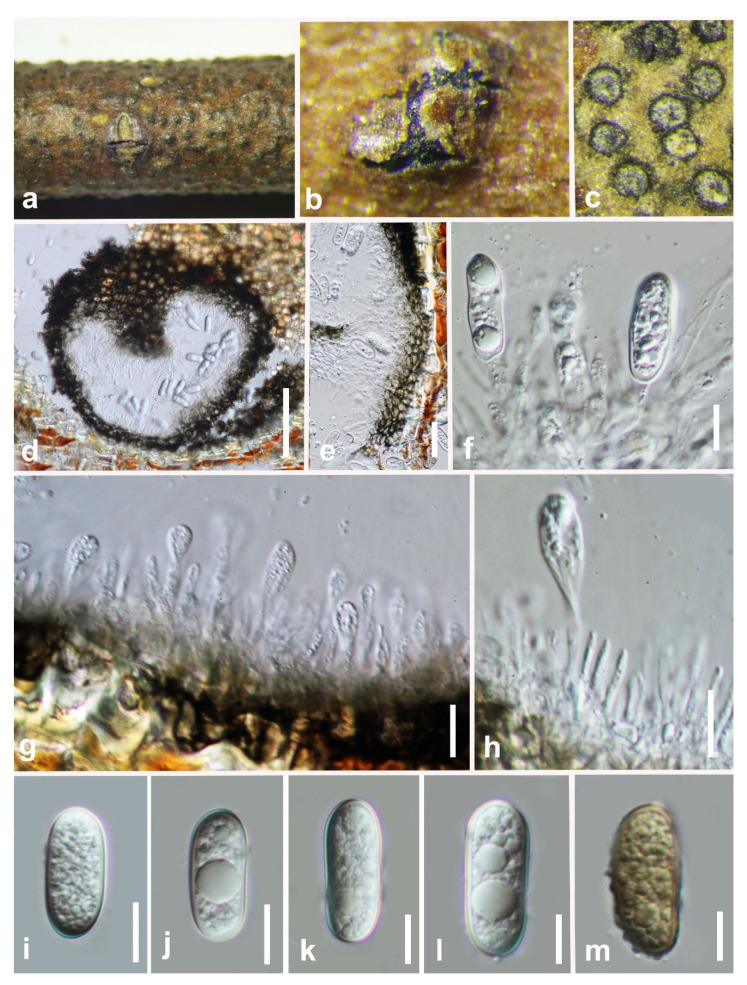
***Sardiniella elliptica*** (GZAAS 19-1838, asexual morph). (**a**–**c**) Conidiomata on host surface. (**d**) Vertical section of conidiomata. (**e**) Peridium. (**f**–**h**) Conidiogenous cells and developing conidia. (**i**–**l**) Immature, hyaline conidia. (**m**) Mature, brown conidia. Scale bars: (**d**) = 100 μm, (**e**) = 20 μm, (**f**–**m**) = 10 μm.

***Sardiniella guizhouensis*** Y.Y. Chen, and Jian K. Liu, Phytotaxa 508: 190 (2021) (Figure 8).

Index Fungorum number: IF558352; Facesoffungi number: FoF09647.

*Saprobic* on decaying wood. **Sexual morph:** Not observed. **Asexual morph:** *Conidiomata* 180–245 × 275–395 µm ($\bar{x}$ = 229 × 330 µm, n = 20), immersed, arranged singly or in small groups within the bark, globose to subglobose, dark brown to black, solitary or gregarious. *Ostiole* central. *Peridium* 22–34 µm thick, outer layer composed of pigmented thick-walled cells of *textura angularis*, inner layer composed of hyaline thin-walled cells of *textura angularis* (three- to five-layered). *Conidiogenous cells* 6–11 × 6–7 µm ($\bar{x}$ = 8.5 × 6.5 μm, n = 25), lining the inner surface of the conidioma, hyaline, short obpyriform to subcylindrical. *Conidia* 21–28 × 10–14 µm ($\bar{x}$ = 24.5 × 12.5 μm, n = 50), ellipsoid to obovoid, immature conidia hyaline, mature conidia becoming medium to dark brown.

**Culture characteristics:** Conidia germinating on WA within 18 h and producing germ tubes from each septum. Colonies growing on PDA, reaching a diameter of 4 cm after five days at 25 °C, effuse, velvety, with entire to slightly undulate edge. The early stage of the white, later green.

**Material examined: China**, Guizhou province, Libo District, Maolan natural reserve, July 2017, GZAAS 19-1948, living culture GZCC 19-0229.

**Notes:***Sardiniella**guizhouensis* was introduced by Chen et al. [37] with both sexual and asexual morphs. One isolate obtained in this study clustered with the ex-type of *Sa.*
*guizhouensis* (CGMCC 3.19222) in the phylogenetic analyses of combined ITS, LSU, and *tef* sequence data (Figure 1). We identified our collection as *Sa*. *guizhouensis* based on morphology and phylogeny.

**Figure 8 jof-07-00893-f008:**
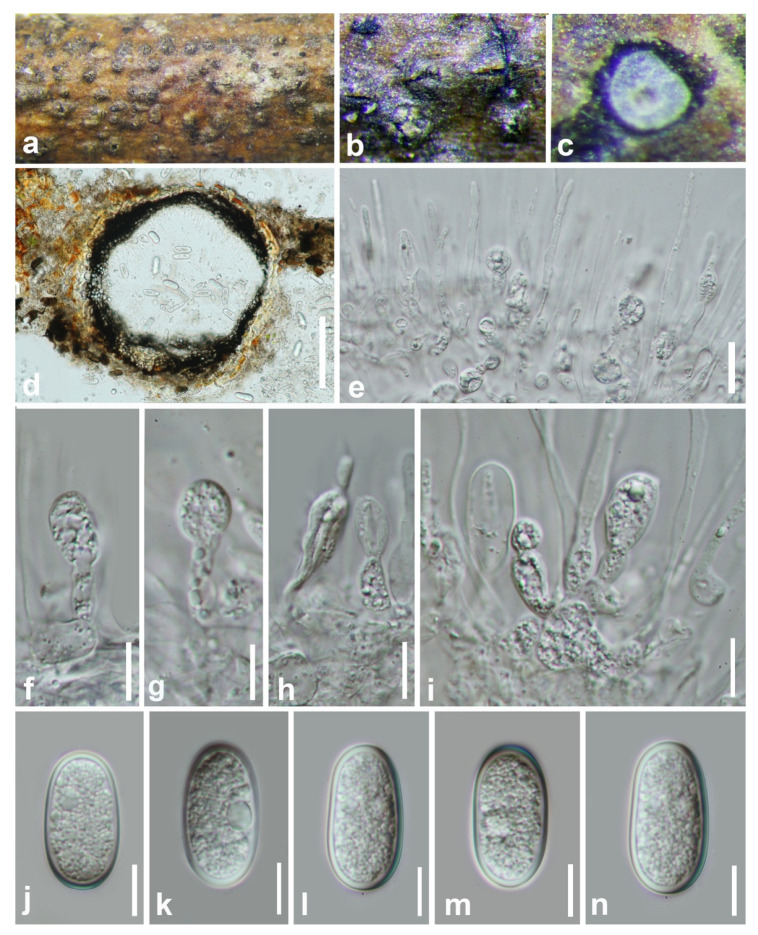
***Sardiniella guizhouensis*** (GZAAS 19-1809, asexual morph). (**a**,**b**) Conidiomata on host surface. (**c**) Horizontal section of conidiomata. (**d**) Vertical section of conidiomata. (**e**–**i**) Conidiogenous cells and developing conidia. (**j**–**n**) Hyaline conidia. Scale bars: (**d**) = 100 μm, (**e**–**n**) = 10 μm.

***Sphaeropsis* *guizhouensis*** Y.Y. Chen, A. J. Dissanayake, and Jian K. Liu., ***sp*. *nov*** (Figure 9).

Index Fungorum number: IF558475; Facesoffungi number: FoF09648.

**Etymology:** Name refers to the location where the fungus was collected, Guizhou, China.

**Holotype:** HKAS 112084.

*Saprobic* on decaying wood. **Sexual morph:** *Ascostromata* 132–185 × 122–165 µm ($\bar{x}$ = 152 × 145 µm, n = 20), initially immersed under host epidermis, becoming semi-immersed to erumpent, breaking through cracks in bark, gregarious and fused, uniloculate, globose to subglobose, membraneous, ostiolate. *Ostiole* 43–52 μm high, 30–42 μm wide, central, papillate, pale brown, relatively broad, periphysate. *Peridium* 28–44 μm wide, broader at the base, comprising several layers of relatively thick-walled, dark brown to black-walled cells arranged in a *textura angularis*. *Pseudoparaphyses* hyphae-like, numerous, embedded in a gelatinous matrix. *Asci* 67–101 × 19–23 µm ($\bar{x}$ = 89 × 20 μm, n = 25), eight-spored, bitunicate, fissitunicate, clavate to cylindro-clavate, sometimes short pedicellate, mostly long pedicellate, apex rounded with an ocular chamber. *Ascospores* 20–23 × 7.8–8.3 µm ($\bar{x}$ = 22 × 8 μm, n = 50), overlapping uniseriate to biseriate, hyaline, aseptate, ellipsoidal to obovoid, slightly wide above the center, minutely guttulate, smooth-walled. **Asexual morph:** Not observed.

**Culture characteristics:** Ascospores germinating on PDA within 18 h. Germ tubes produced from both ends of the ascospores. Fast growing; fimbriate, flat or effuse, dense, convex with papillate surface, reaching the edge of the Petri dish after seven days.

**Material examined: China**, Guizhou province, Libo District, Maolan natural reserve, on decaying woody host, July 2017, Y.Y. Chen, GZAAS4 (HKAS 112084, holotype); ex-type living culture CGMCC 3.20352; *ibid.*, Xingyi Wanfenglin, on decaying woody host, June 2019, Y.Y. Chen, (GZAAS 19-2892, paratype), living culture GZCC 19-0273.

**Notes:***Sphaeropsis guizhouensis* formed a distinct clade (Figure 1) and is phylogenetically distinct from *Sp. eucalypticola* (MFLUCC 11-0579) in a clade with absolute support (ML/MP/BI = 100/100/1.0). *Sphaeropsis guizhouensis* can be distinguished from *Sp. eucalypticola* based on ITS (3/680 bp), LSU (1/803 bp), and *tef* (25/479 bp). *Sphaeropsis guizhouensis* differs from *Sp. eucalypticola* in having smaller asci (89 × 20 μm vs. 106 × 29 μm) and ascospores (22 × 8 μm vs. 30 × 12 μm).

**Figure 9 jof-07-00893-f009:**
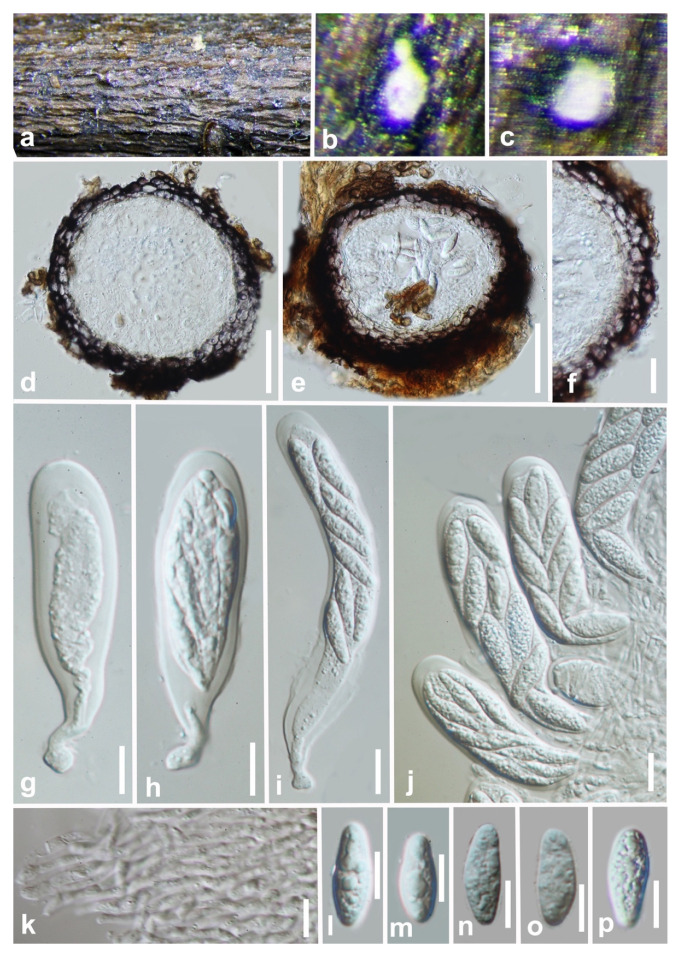
***Sphaeropsis guizhouensis*** (HKAS 112084, holotype). (**a**–**c**) Appearance of ascomata on wood. (**d**,**e**) Vertical section of ascomata. (**f**) Peridium. (**g**–**j**) Immature and mature asci. (**k**) Pseudoparaphyses. (**l**–**p**) Hyaline mature ascospores. Scale bars: (**d**,**e**) = 50 μm, (**f**) = 20 μm, (**g**–**p**) = 10 μm.

**Figure 10 jof-07-00893-f010:**
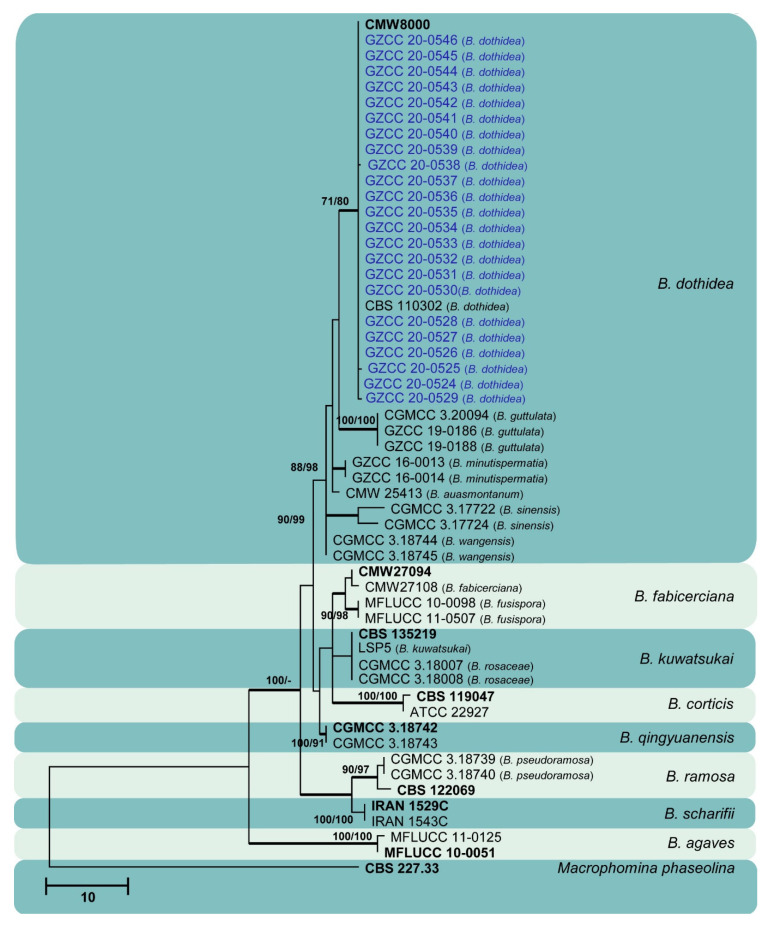
Maximum likelihood tree based on analysis of combined ITS and *tef* sequences of *Botryosphaeria*. Bootstrap support values of ML, MP > 75% are shown near the nodes, and branches in bold indicate BI probabilities >0.95. Isolates obtained in this study are in blue for known species. Ex-type strains are in bold. The tree is rooted to *Macrophomina phaseolina* (CBS 227.33).

**Figure 11 jof-07-00893-f011:**
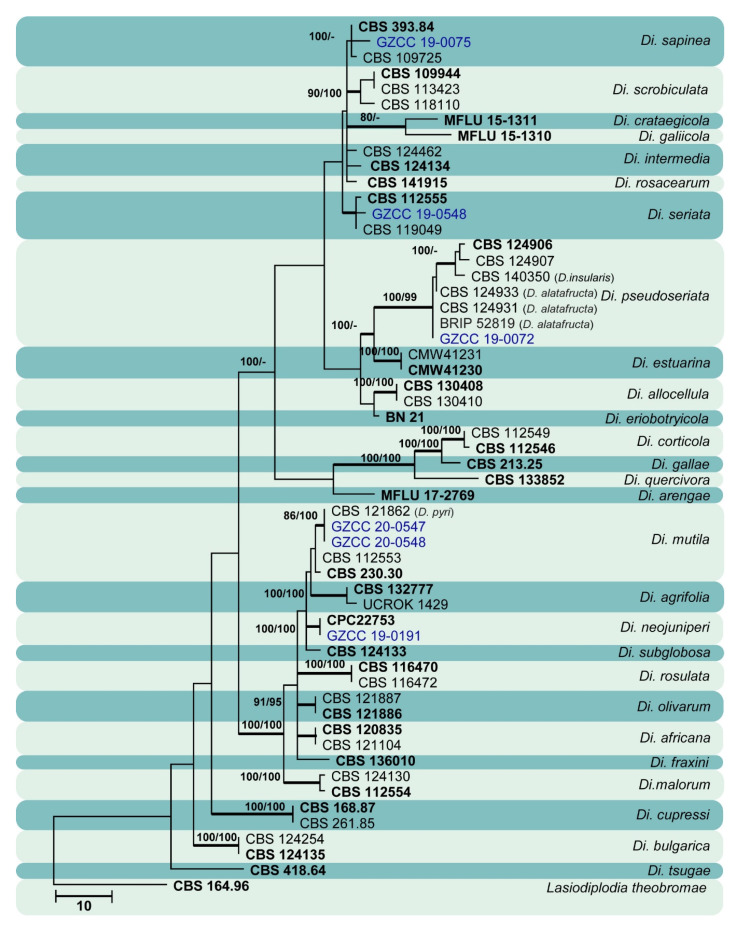
Maximum likelihood tree based on analysis of combined ITS and *tef* sequences of *Diplodia*. Bootstrap support values of ML, MP > 75% are shown near the nodes, and branches in bold indicate BI probabilities >0.95. Isolates obtained in this study are in blue for known species. Ex-type strains are in bold. The tree is rooted to *Lasiodiplodia theobromae* (CBS 164.96).

**Figure 12 jof-07-00893-f012:**
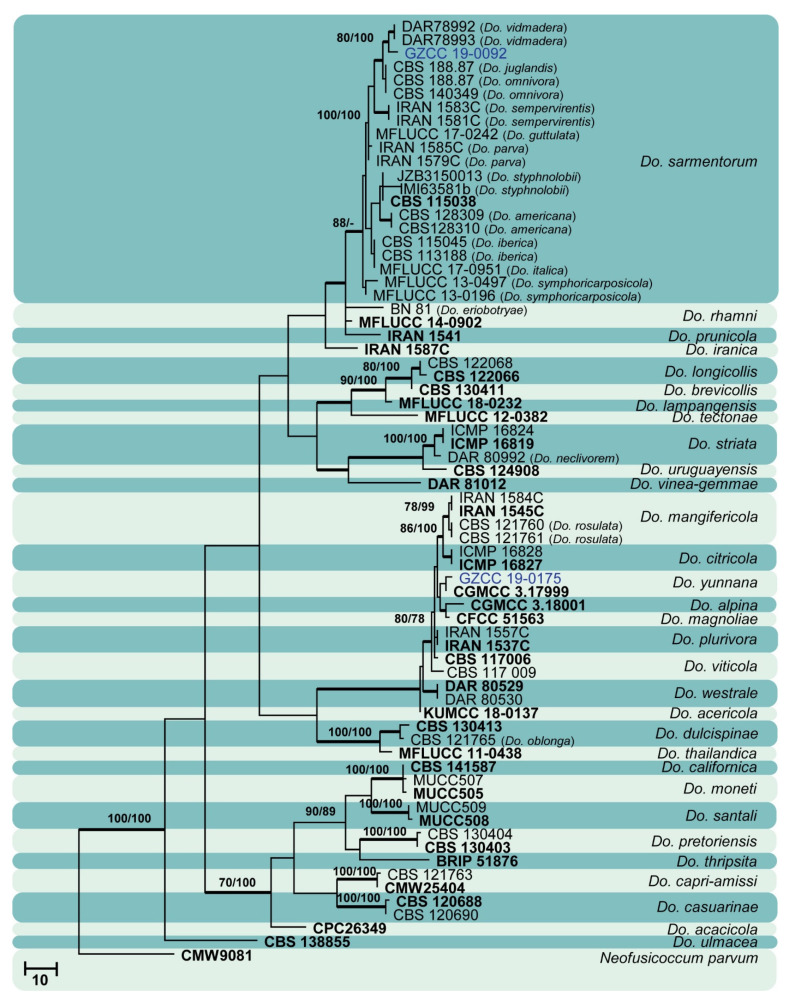
Maximum likelihood tree based on analysis of combined ITS and *tef* sequences of *Dothiorella*. Bootstrap support values of ML, MP > 75% are shown near the nodes, and branches in bold indicate BI probabilities >0.95. Isolates obtained in this study are in blue for known species. Ex-type strains are in bold. The tree is rooted to *Neofusicoccum parvum* (CMW9081).

**Figure 13 jof-07-00893-f013:**
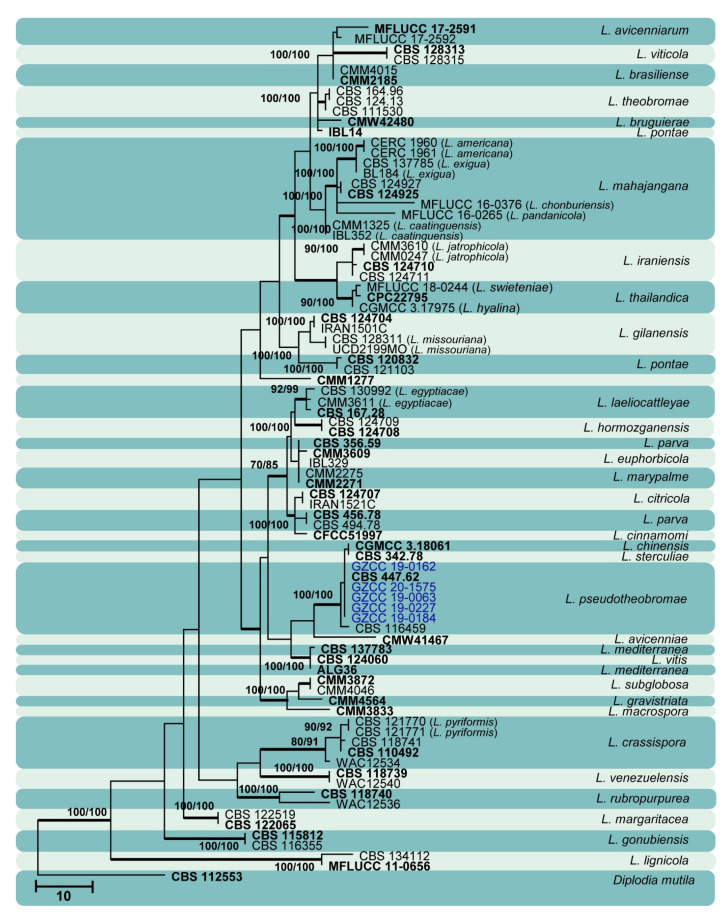
Maximum likelihood tree based on analysis of combined ITS and *tef* sequences of *Lasiodiplodia*. Bootstrap support values of ML, MP > 75% are shown near the nodes, and branches in bold indicate BI probabilities >0.95. Isolates obtained in this study are in blue for known species. Ex-type strains are in bold. The tree is rooted to *Diplodia mutila* (CBS 112553).

**Figure 14 jof-07-00893-f014:**
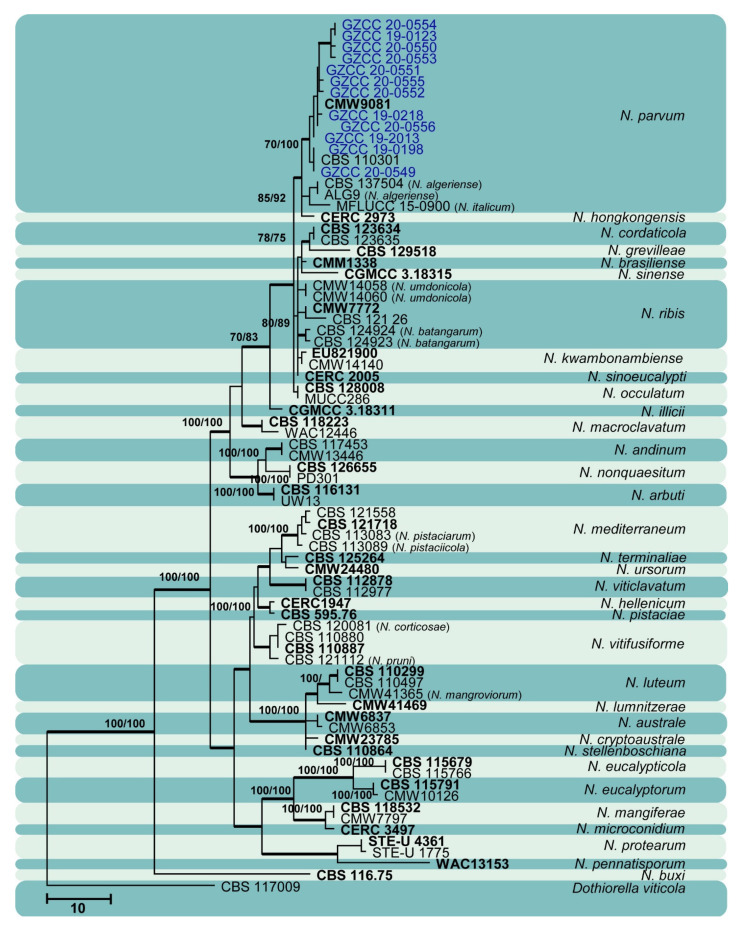
Maximum likelihood tree based on analysis of combined ITS and *tef* sequences of *Neofusicoccum*. Bootstrap support values of ML, MP > 75% are shown near the nodes, and branches in bold indicate BI probabilities >0.95. Isolates obtained in this study are in blue for known species. Ex-type strains are in bold. The tree is rooted to *Dothiorella viticola* (CBS 117009).

## 4. Discussion

This study reports the largest collection of *Botryosphaeriales* isolates from Guizhou province, China, and is the first attempt to characterize *Botryosphaeriales* species from various nature reserves in the province. Three new species were described in three rarely observed genera: *Botryobambusa* (*Bo. guizhouensis*), *Sardiniella* (*Sa. elliptica*), and *Sphaeropsis* (*Sp. guizhouensis*). These three genera have few species and have been sporadically isolated worldwide. Apart from these novel species, 15 known species—*Aplosporella hesperidica*, *Barriopsis tectonae*, *Botryosphaeria dothidea, Diplodia mutila*, *Di. neojuniperi, Di. pseudoseriata*, *Di*. *sapinea*, *Di*. *seriata*, *Dothiorella sarmentorum*, *Do*. *yunnana*, *Lasiodiplodia pseudotheobromae*, *Neofusicoccum parvum*, *Sardiniella celtidis*, *Sa. guizhouensis*, and *Sphaeropsis citrigena*—were isolated and included in their respective phylogenies.

The genus *Aplosporella* consists of plant pathogens, saprobes, and endophytes. Though 300 epithets of *Aplosporella* are registered in Index Fungorum (http://www.indexfungorum.org/Names/Names.asp, accessed in 23 August 2021), only 12 species are accepted within this genus (www.botryosphaeriales.org). *Aplosporella hesperidica* has been reported in several Asian countries [17,22,38,39,40,41,42], but this is the first time it has been identified in China.

Five species are accepted in the genus *Barriopsis* [35,43,44,45]. *Barriopsis tectonae* was introduced by Doilom et al. [35] from a dead *Tectona grandis* branch collected in Thailand. So far, this species has been reported only from Thailand (http://nt.arsgrin.gov/fungaldatabases/, accessed on 25 September 2021), and here we provide a new country report of *Ba. tectonae* (Figure 3) based on sexual morphological characteristics and molecular evidence.

*Botryobambusa* is a monotypic genus (www.botryosphaeriales.org). Liu et al. [1] introduced and compared *Botryobambusa* with other existing genera in *Botryosphaeriales*. It is thus far only identified from bamboo in Thailand. In this study, two isolates obtained from decaying bamboo in Forest Park, Chishui District in Guizhou province were identified as a novel species (*Bo. guizhouensis*). The sexual morph of *Bo. guizhouensis* was distinguished from *Bo. fusicoccum* by its larger asci and ascospores as well as by molecular phylogeny. Our study shows that the genus can be clearly discriminated from the morphologically similar genus *Botryosphaeria* by its velvety, hyaline, and sheathed ascospores. Phylogenetically, these two genera are clearly distinct lineages.

Linaldeddu et al. [46] introduced the genus *Sardiniella* by denoting *Sa. urbana* as the type species. Hyde et al. [36] introduced the second species in the genus, *Sa. celtidis*, while Chen et al. [37] introduced the third species, *Sa. guizhouensis*, reporting a sexual morph of the genus for the first time. In this study, another new species (*Sa. elliptica*) is described and assigned to the genus with details provided for a previously known species (*Sa. celtidis*). With morphological and molecular support, here we present the sexual morph report for *Sa. elliptica*; a newly introduced species in this study (Figure 6 and Figure 7). So far, *Sardiniella* species are known only from Italy and China (http://nt.arsgrin.gov/fungaldatabases/, accessed on 25 September 2021).

Though more than 630 names exist in *Sphaeropsis* (Index Fungorum, August 2021), only five species are currently accepted [2,3]. In this study, two isolates obtained from decaying woody hosts in Guizhou province were identified as a novel species, *Sp. guizhouensis*. The sexual morph of *Sp. guizhouensis* (Figure 9) is distinguished from the other species in this genus by ascospore dimensions. Another previously known species *Sp. citrigena* was also isolated and included in the phylogenetic analysis.

*Botryosphaeria dothidea* (Figure 10) and *Neofusicoccum parvum* (Figure 14) were the most isolated species in this study, consistent with some prior studies [47,48,49,50,51], which indicates the ability of species in these genera to inhabit a variety of plant species and geographic areas globally. Certain *Diplodia* species occupy extensive host ranges, such as *Di. seriata* which has been documented on more than 250 hosts [3,5]. In this study, we isolated five *Diplodia* species (Figure 11): *Di*. *mutila*, *Di. neojuniperi*, *Di. pseudoseriata*, *Di*. *sapinea*, and *Di*. *seriata*. Our study revealed two previously known *Dothiorella* species (Figure 12), *Do. sarmentorum* and *Do*. *yunnana*, for the first time from Guizhou province. *Lasiodiplodia pseudotheobromae* is a common and cosmopolitan species on diverse host plants and has been reported from various localities globally. This study revealed five saprobic *L. pseudotheobromae* isolates (Figure 13) in Guizhou province.

Members of *Botryosphaeriales* signify a rising risk to agricultural crops and urban and natural forest ecosystems in China. Collecting and identifying *Botryosphaeriales* isolates from various hosts and locations is required to describe and understand these species. The occurrence and significance of *Botryosphaeriales* species in various nature reserves has not been investigated at a larger scale so far in Guizhou province. Hence, in this study, we provided a larger collection of *Botryosphaeriales* isolates and identify them to species level by both morphology and phylogeny. Further studies are needed to explore and gather data on their occurrence, as precise data of the causal agents is essential.

## 5. Conclusions

We carried out fungal diversity investigations at a large scale in southwestern China and here we provided a report of *Botryosphaeriales* species isolated from various woody hosts in Guizhou province, China. The identification of 18 *Botryosphaeriales* species (15 known species and three new species) associated with saprobic woody hosts was revealed.

## Figures and Tables

**Figure 1 jof-07-00893-f001:**
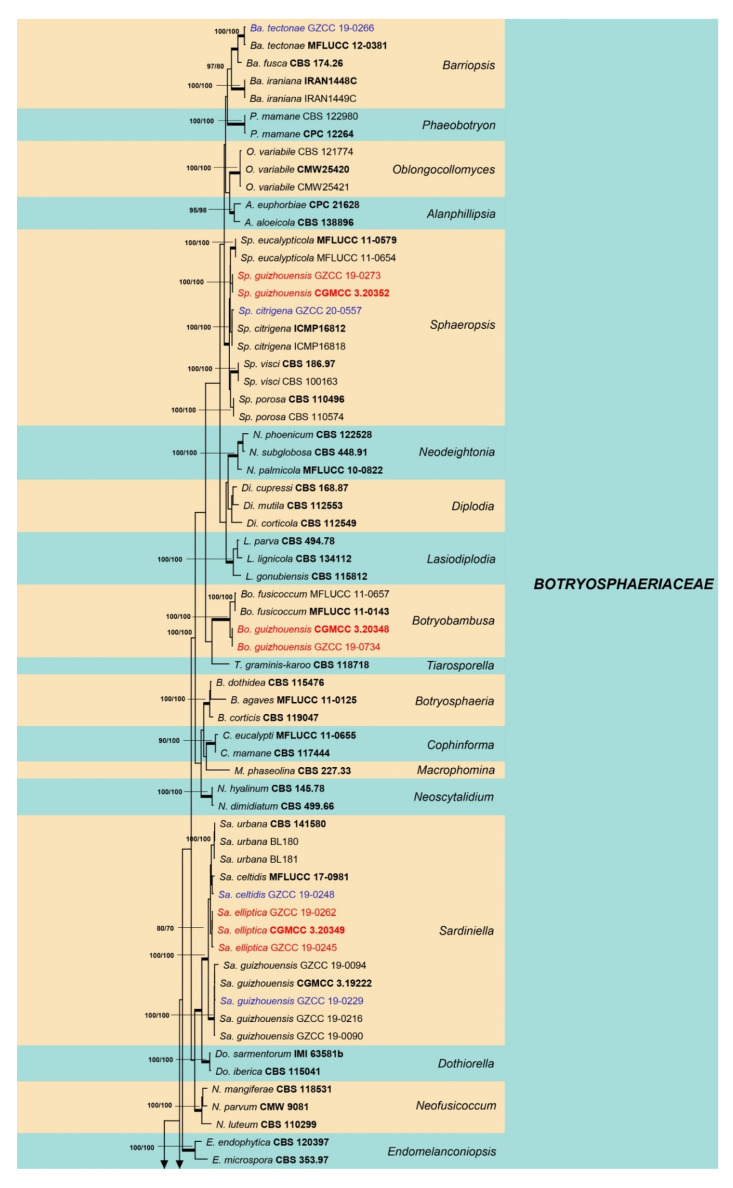
Maximum likelihood tree based on analysis of combined ITS, LSU, and *tef* sequences of selected *Botryosphaeriales* taxa. Bootstrap values of ML, MP >75% are shown near nodes and branches in bold indicate BI probabilities >0.95. Isolates obtained in this study are in blue for known species, and novel taxa are in red. Ex-type strains are in bold. The tree is rooted to *Lecanosticta acicola* (LNPV252). The scale bar represents the expected number of nucleotide substitutions per site.

**Table 1 jof-07-00893-t001:** *Botryosphaeriales* species isolated and characterized in this study. The details of the type species of novel taxa are given in bold. N/A: No sequence available.

Species Name	Isolate Number	Location	Date	ITS	LSU	*tef*
*Aplosporella hesperidica*	GZCC 19-0095	Fanjing mountain, Tongren District	July 2018	MZ781423	MZ781490	MZ852496
*Barriopsis tectonae*	GZCC 19-0266	Maolan natural reserve, Libo District	July 2017	MZ781424	MZ781491	MZ852497
** *Botryobambusa guizhouensis* **	**CGMCC 3. 20348**	**Forest Park, Chishui District**	**July 2019**	**MZ781425**	**MZ781492**	**MZ852498**
	GZCC 19-0734	Forest Park, Chishui District	July 2019	MZ781426	MZ781493	MZ852499
*Botryosphaeria dothidea*	GZCC 20-0524	Huaxi wetland park, Guiyang District	April 2017	MZ781435	N/A	MZ852508
	GZCC 20-0525	Xingyi Wanfenglin	June 2019	MZ781436	N/A	MZ852509
	GZCC 20-0526	Xingyi Wanfenglin	June 2019	MZ781437	N/A	MZ852510
	GZCC 20-0527	Suiyang broad water nature reserve	April 2018	MZ781438	N/A	MZ852511
	GZCC 20-0528	Forest Park, Chishui District	July 2019	MZ781439	N/A	MZ852512
	GZCC 20-0529	Suiyang broad water nature reserve	April 2018	MZ781440	N/A	MZ852513
	GZCC 20-0530	Suiyang broad water nature reserve	April 2018	MZ781441	N/A	MZ852514
	GZCC 20-0531	Xiaochehe wetland park, Guiyang District	May 2017	MZ781442	N/A	MZ852515
	GZCC 20-0532	Forest Park, Chishui District	July 2019	MZ781443	N/A	MZ852516
	GZCC 20-0533	Huaxi wetland park, Guiyang District	April 2017	MZ781444	N/A	MZ852517
	GZCC 20-0534	Suiyang broad water nature reserve	April 2018	MZ781445	N/A	MZ852518
	GZCC 20-0535	Xiaochehe wetland park, Guiyang District	May 2017	MZ781446	N/A	MZ852519
	GZCC 20-0536	Xiaochehe wetland park, Guiyang District	May 2017	MZ781447	N/A	MZ852520
	GZCC 20-0537	Huaxi wetland park, Guiyang District	April 2017	MZ781448	N/A	MZ852521
	GZCC 20-0538	Forest Park, Chishui District	July 2019	MZ781449	N/A	MZ852522
	GZCC 20-0539	Forest Park, Chishui District	July 2019	MZ781450	N/A	MZ852523
	GZCC 20-0540	Huaxi wetland park, Guiyang District	April 2017	MZ781451	N/A	MZ852524
	GZCC 20-0541	Forest Park, Chishui District	July 2019	MZ781452	N/A	MZ852525
	GZCC 20-0542	Forest Park, Chishui District	July 2019	MZ781453	N/A	MZ852526
	GZCC 20-0543	Xiaochehe wetland park, Guiyang District	May 2017	MZ781454	N/A	MZ852527
	GZCC 20-0544	Suiyang broad water nature reserve	April 2018	MZ781455	N/A	MZ852528
	GZCC 20-0545	Xingyi Wanfenglin	June 2019	MZ781456	N/A	MZ852529
	GZCC 20-0546	Xiaochehe wetland park, Guiyang District	May 2017	MZ781457	N/A	MZ852530
*Diplodia mutila*	GZCC 20-0547	Xingyi Wanfenglin	June 2019	MZ781459	N/A	MZ852531
	GZCC 20-0548	Huaxi wetland park, Guiyang District	April 2017	MZ781460	N/A	MZ852532
*Di. neojuniperi*	GZCC 19-0191	Maolan natural reserve, Libo District	July 2017	MZ781463	N/A	MZ852533
*Di. pseudoseriata*	GZCC 19-0072	Xingyi Wanfenglin	June 2019	MZ781461	N/A	MZ852534
*Di*. *sapinea*	GZCC 19-0075	Suiyang broad water nature reserve	April 2018	MZ781462	N/A	MZ852535
*Di*. *seriata*	GZCC 19-0548	Xiaochehe wetland park, Guiyang District	May 2017	MZ781458	N/A	MZ852536
*Dothiorella sarmentorum*	GZCC 19-0092	Xingyi Wanfenglin	June 2019	MZ781464	N/A	MZ852537
*Do*. *yunnana*	GZCC 19-0175	Suiyang broad water nature reserve	April 2018	MZ781465	N/A	MZ852538
*Lasiodiplodia pseudotheobromae*	GZCC 19-0162	Xiaochehe wetland park, Guiyang District	May 2017	MZ781466	N/A	MZ852539
	GZCC 19-0184	Xiaochehe wetland park, Guiyang District	May 2017	MZ781467	N/A	MZ852540
	GZCC 19-0227	Huaxi wetland park, Guiyang District	April 2017	MZ781468	N/A	MZ852541
	GZCC 20-1575	Xiaochehe wetland park, Guiyang District	May 2017	MZ781469	N/A	MZ852542
	GZCC 19-0063	Suiyang broad water nature reserve	April 2018	MZ781470	N/A	MZ852543
*Neofusicoccum parvum*	GZCC 19-0123	Xiaochehe wetland park, Guiyang District	May 2017	MZ781471	N/A	MZ852544
	GZCC 19-0198	Xingyi Wanfenglin	June 2019	MZ781472	N/A	MZ852545
	GZCC 19-0218	Xingyi Wanfenglin	June 2019	MZ781473	N/A	MZ852546
	GZCC 19-2013	Maolan natural reserve, Libo District	July 2017	MZ781474	N/A	MZ852547
	GZCC 20-0549	Xiaochehe wetland park, Guiyang District	May 2017	MZ781475	N/A	MZ852548
	GZCC 20-0550	Xiaochehe wetland park, Guiyang District	May 2017	MZ781476	N/A	MZ852549
	GZCC 20-0551	Xingyi Wanfenglin	June 2019	MZ781477	N/A	MZ852550
	GZCC 20-0552	Xiaochehe wetland park, Guiyang District	May 2017	MZ781478	N/A	MZ852551
	GZCC 20-0553	Huaxi wetland park, Guiyang District	April 2017	MZ781479	N/A	MZ852552
	GZCC 20-0554	Xiaochehe wetland park, Guiyang District	May 2017	MZ781480	N/A	MZ852553
	GZCC 20-0555	Suiyang broad water nature reserve	April 2018	MZ781481	N/A	MZ852554
	GZCC 20-0556	Xiaochehe wetland park, Guiyang District	May 2017	MZ781482	N/A	MZ852555
*Sardiniella celtidis*	GZCC 19-0248	Xingyi Wanfenglin	June 2019	MZ781427	MZ781494	MZ852500
** *Sa. elliptica* **	**CGMCC 3.20349**	**Huaxi wetland park, Guiyang District**	**April 2017**	**MZ781429**	**MZ781496**	**MZ852502**
	GZCC 19-0245	Xingyi Wanfenglin	June 2019	MZ781431	MZ781498	MZ852504
	GZCC 19-0262	Maolan natural reserve, Libo District	July 2017	MZ781430	MZ781497	MZ852503
*Sa. guizhouensis*	GZCC 19-0229	Maolan natural reserve, Libo District	July 2017	MZ781428	MZ781495	MZ852501
*Sphaeropsis citrigena*	GZCC 20-0557	Xingyi Wanfenglin	June 2019	MZ781432	MZ781499	MZ852505
** *Sp. guizhouensis* **	**CGMCC 3.20352**	**Maolan natural reserve** **, Libo District**	**July 2017**	**MZ781433**	**MZ781500**	**MZ852506**
	GZCC 19-0273	Xingyi Wanfenglin	June 2019	MZ781434	MZ781501	MZ852507

**Table 2 jof-07-00893-t002:** Details of gene regions, respective primer pairs, primer sequences, and PCR conditions used in the study.

Gene Region	Primers	Sequence 5′-3′	Optimized PCR Protocols	References
ITS	ITS1	TCCGTAGGTGAACCTGCGG	94 °C: 5 min, (94 °C: 30 s, 55 °C: 50 s, 72 °C: 1 min) × 34 cycles 72 °C: 10 min	White et al. [26]
	ITS4	TCCTCCGCTTATTGATATGC		
LSU	LR0R	ACCCGCTGAACTTAAGC	94 °C: 5 min, (94 °C: 30 s, 54 °C: 50 s, 72 °C: 1 min) × 34 cycles 72 °C: 10 min	Vilgalys and Hester [27]
	LR5	TCCTGAGGGAAACTTCG		
*tef*	EF1-728F	CATCGAGAAGTTCGAGAAGG	95 °C: 5 min, (95 °C: 30 s, 58 °C: 30 s, 72 °C: 1 min) × 34 cycles 72 °C: 10 min	Carbone and Kohn [28]
	EF1-986R	TACTTGAAGGAACCCTTACC		

**Table 3 jof-07-00893-t003:** *Botryosphaeriales* taxa used in the phylogenetic analysis. The culture collection accession number for the type strains are given in bold. N/A: No sequence available.

Species Name	Isolate Number	ITS	LSU	*tef*
*Aplosporellaceae*				
*Alanomyces indica*	**CBS 134264**	HF563622	HF563623	AB872219
*Aplosporella africana*	**CBS 121777**	KF766196	EU101380	EU101360
*A. africana*	**CBS 121778**	EU101316	EU101381	EU101361
*A. artocarpi*	**CPC 22791**	KM006450	N/A	KM006481
*A. ginkgonis*	**CFCC 52442**	MH133916	MH133933	MH133950
*A. ginkgonis*	CFCC 89661	KM030583	KM030590	KM030597
*A. hesperidica*	**CBS 732.79**	KX464083	KX464239	N/A
*A. hesperidica*	CBS 208.37	JX681069	MH867398	N/A
*A. javeedii*	**CFCC 50054**	KP208840	KP208843	KP208846
*A. javeedii*	CFCC 50052	KP208838	KP208841	KP208844
*A. macropycnidia*	**CGMCC 3.17725**	KT343648	N/A	KX011176
*A. macropycnidia*	CGMCC 3.17726	KT343649	N/A	KX011177
*A. papillata*	**CBS 121780**	EU101328	EU101383	EU101373
*A. papillata*	CBS 121781	EU101329	EU101384	EU101374
*A. prunicola*	**CBS 121167**	KF766147	KF766315	N/A
*A. prunicola*	STE-U 6326	EF564375	EF564377	N/A
*A. sophorae*	**CPC 29688**	KY173388	KY173482	N/A
*A. thailandica*	**MFLU 16-0615**	KX423536	N/A	KX423537
*A. yalgorensis*	**MUCC511**	EF591926	EF591943	EF591977
*A. yalgorensis*	MUCC512	EF591927	EF591944	EF591978
*Botryosphaeriaceae*				
*Alanphillipsia aloeicola*	**CBS 138896**	KP004444	KP004472	N/A
*A. euphorbiae*	**CPC 21628**	KF777140	KF777196	N/A
*Barriopsis fusca*	**CBS 174.26**	EU673330	DQ377857	EU673296
*Ba. iraniana*	**IRAN1448C**	FJ919663	KF766318	FJ919652
*Ba. iraniana*	IRAN1449C	FJ919665	N/A	FJ919654
*Ba. tectonae*	**MFLUCC 12-0381**	KJ556515	N/A	KJ556516
*Botryobambusa fusicoccum*	**MFLUCC 11-0143**	JX646792	JX646809	JX646857
*Bo. fusicoccum*	MFLUCC 11-0657	JX646793	JX646810	JX646858
*Botryosphaeria agaves*	**MFLUCC 11-0125**	JX646791	JX646808	JX646856
*B. agaves*	MFLUCC 10-0051	JX646790	JX646807	JX646855
*B. corticis*	**CBS 119047**	DQ299245	EU673244	EU017539
*B. corticis*	ATCC 22927	DQ299247	EU673245	EU673291
*B. dothidea*	**CMW8000**	AY236949	DQ377852	AY236898
*B. dothidea*	CBS 110302	AY259092	EU673243	AY573218
*B. dothidea*	CMW 25413	KF766167	KF766332	EU101348
*B. dothidea*	GZCC 16-0013	KX447675	N/A	KX447678
*B. dothidea*	GZCC 16-0014	KX447676	N/A	KX447679
*B. dothidea*	CGMCC 3.17722	KT343254	N/A	KU221233
*B. dothidea*	CGMCC 3.17724	KT343256	N/A	KU221234
*B. dothidea*	CGMCC 3.18744	KX278002	N/A	KX278107
*B. dothidea*	CGMCC 3.18745	KX278003	N/A	KX278108
*B. dothidea*	CGMCC3.20094	MT327839	N/A	MT331606
*B. dothidea*	GZCC 19-0186	MT327832	N/A	MT331600
*B. dothidea*	GZCC 19-0188	MT327833	N/A	MT331601
*B. fabicerciana*	**CMW27094**	HQ332197	N/A	HQ332213
*B. fabicerciana*	CMW27108	HQ332200	N/A	HQ332216
*B. fabicerciana*	MFLUCC 10-0098	JX646789	JX646806	JX646854
*B. fabicerciana*	MFLUCC 11-0507	JX646788	JX646805	JX646853
*B. kuwatsukai*	**CBS 135219**	KJ433388	N/A	KJ433410
*B. kuwatsukai*	LSP5	KJ433395	N/A	KJ433417
*B. kuwatsukai*	CGMCC 3.18007	KX197074	N/A	KX197094
*B. kuwatsukai*	CGMCC 3.18008	KX197075	N/A	KX197095
*B. qingyuanensis*	**CGMCC 3.18742**	KX278000	N/A	KX278105
*B. qingyuanensis*	CGMCC 3.18743	KX278001	N/A	KX278106
*B. ramosa*	**CBS 122069**	EU144055	N/A	EU144070
*B. ramosa*	CGMCC 3.18739	KX277988	N/A	KX278093
*B. ramosa*	CGMCC 3.18740	KX277989	N/A	KX278094
*B. scharifii*	**IRAN1529C**	JQ772020	N/A	JQ772057
*B. scharifii*	IRAN1543C	JQ772019	N/A	JQ772056
*Cophinforma eucalypti*	**MFLUCC 11-0655**	JX646801	JX646818	JX646866
*C. mamane*	**CBS 117444**	KF531822	DQ377855	KF531801
*Diplodia africana*	**CBS 120835**	EF445343	N/A	EF445382
*Di. africana*	CBS 121104	EF445344	N/A	EF445383
*Di. agrifolia*	**CBS 132777**	JN693507	N/A	JQ517317
*Di. agrifolia*	UCROK1429	JQ411412	N/A	JQ512121
*Di. allocellula*	**CBS 130408**	JQ239397	JQ239410	JQ239384
*Di. allocellula*	CBS 130410	JQ239399	JQ239412	JQ239386
*Di. arengae*	**MFLU 17-2769**	MG762771	N/A	MG762774
*Di. bulgarica*	**CBS 124135**	GQ923853	N/A	GQ923821
*Di. bulgarica*	CBS 124254	GQ923852	N/A	GQ923820
*Di. corticola*	**CBS 112546**	AY259100	AY928051	AY573227
*Di. corticola*	CBS 112549	AY259110	EU673262	DQ458872
*Di. crataegicola*	**MFLU 15-1311**	KT290244	N/A	KT290248
*Di. cupressi*	**CBS 168.87**	DQ458893	EU673263	DQ458878
*Di. cupressi*	CBS 261.85	DQ458894	EU673264	DQ458879
*Di. eriobotryicola*	**BN 21**	MT587342	N/A	MT592047
*Di. estuarina*	**CMW41231**	KP860831	N/A	KP860676
*Di. estuarina*	CMW41230	KP860830	N/A	KP860675
*Di. fraxini*	**CBS 136010**	KF307700	N/A	KF318747
*Di. galiicola*	**MFLU 15-1310**	KT290245	N/A	KT290249
*Di. gallae*	**CBS 213.25**	KX464092	N/A	KX464566
*Di. malorum*	**CBS 124130**	GQ923865	N/A	GQ923833
*Di. malorum*	CBS 112554	AY259095	N/A	DQ458870
*Di. mutila*	**CBS 112553**	AY259093	AY928049	AY573219
*Di. mutila*	CBS 230.30	DQ458886	AY928049	DQ458869
*Di. mutila*	CBS 121862	KX464093	N/A	KX464567
*Di. neojuniperi*	**CPC22753**	KM006431	N/A	KM006462
*Di. olivarum*	**CBS 121887**	EU392302	N/A	EU392279
*Di. olivarum*	CBS 121886	EU392297	N/A	EU392274
*Di. pseudoseriata*	**CBS 124906**	EU080927	MH874931	EU863181
*Di. pseudoseriata*	CBS 124907	EU080922	N/A	EU863179
*Di. pseudoseriata*	CBS 124931	FJ888460	MH874935	FJ888444
*Di. pseudoseriata*	CBS 124933	FJ888478	N/A	FJ888446
*Di. pseudoseriata*	CBS 140350	KX833072	N/A	KX833073
*Di. quercivora*	**CBS 133852**	JX894205	N/A	JX894229
*Di. rosulata*	**CBS 116470**	EU430265	DQ377896	EU430267
*Di. rosulata*	CBS 116472	EU430266	DQ377897	EU430268
*Di. sapinea*	**CBS 393.84**	DQ458895	DQ377893	DQ458880
*Di. sapinea*	CBS 109725	DQ458896	EU673270	DQ458881
*Di. sapinea*	CBS 124462	GQ923858	MH874896	GQ923826
*Di. sapinea*	CBS 124134	HM036528	N/A	GQ923851
*Di. sapinea*	CBS 141915	KT956270	N/A	KU378605
*Di. scrobiculata*	**CBS 118110**	AY253292	KF766326	AY624253
*Di. scrobiculata*	CBS 109944	DQ458899	EU673268	DQ458884
*Di. scrobiculata*	CBS 113423	DQ458900	EU673267	DQ458885
*Di. seriata*	**CBS 112555**	AY259094	AY928050	AY573220
*Di. seriata*	CBS 119049	DQ458889	EU673266	DQ458874
*Di. subglobosa*	**CBS 124133**	GQ923856	N/A	GQ923824
*Di. tsugae*	**CBS 418.64**	DQ458888	DQ377867	DQ458873
*Dothiorella acacicola*	**CPC26349**	KX228269	KX228320	KX228376
*Do. acericola*	**KUMCC 18-0137**	KY385661	N/A	KY393212
*Do. alpina*	**CGMCC 3.18001**	KX499645	N/A	KX499651
*Do. brevicollis*	**CBS 130411**	JQ239403	JQ239416	JQ239390
*Do. capri-amissi*	**CBS 121763**	EU101323	N/A	EU101368
*Do. capri-amissi*	CMW25404	EU101324	N/A	EU101369
*Do. casuarinae*	**CBS 120688**	DQ846773	N/A	DQ875331
*Do. casuarinae*	CBS 120690	DQ846774	N/A	DQ875333
*Do. citricola*	**ICMP 16828**	EU673323	N/A	EU673290
*Do. citricola*	ICMP 16827	EU673322	N/A	EU673289
*Do. dulcispinae*	**CBS 130413**	JQ239400	JQ239413	JQ239387
*Do. dulcispinae*	CBS 121765	EU101300	KX464317	EU101345
*Do. iranica*	**IRAN1587C**	KC898231	N/A	KC898214
*Do. juglandis*	**CBS 188.87**	EU673316	DQ377891	EU673283
*Do. lampangensis*	**MFLUCC 18-0232**	MK347758	N/A	MK340869
*Do. longicollis*	**CBS 122068**	EU144054	MH874718	EU144069
*Do. longicollis*	CBS 122066	EU144052	KX464311	EU144067
*Do. magnoliae*	**CFCC 51563**	KY111248	N/A	KY213687
*Do. mangifericola*	**IRAN1584C**	MT587407	N/A	MT592119
*Do. mangifericola*	IRAN1545C	KC898221	N/A	KX464614
*Do. mangifericola*	CBS 121760	KF766227	N/A	EU101335
*Do. mangifericola*	CBS 121761	EU101293	N/A	EU101338
*Do. moneti*	**MUCC 505**	EF591920	EF591937	EF591971
*Do. moneti*	MUCC 507	EF591922	EF591939	EF591973
*Do. plurivora*	**IRAN1557C**	KC898225	N/A	KC898208
*Do. plurivora*	IRAN1537C	KC898226	N/A	KC898209
*Do. pretoriensis*	**CBS 130404**	JQ239405	JQ239418	JQ239392
*Do. pretoriensis*	CBS 130403	JQ239406	JQ239419	JQ239393
*Do. prunicola*	**IRAN1541**	EU673313	EU673232	EU673280
*Do. rhamni*	**MFLUCC 14-0902**	KU246381	KU246382	N/A
*Do. rhamni*	BN 81	MT587399	N/A	MT592111
*Do. santali*	**MUCC 509**	EF591924	EF591941	EF591975
*Do. santali*	MUCC 508	EF591923	EF591940	EF591974
*Do. sarmentorum*	**IMI63581b**	AY573212	AY928052	AY573235
*Do. sarmentorum*	CBS 115038	AY573206	DQ377860	AY573223
*Do. sarmentorum*	CBS 128309	HQ288218	MH876298	HQ288262
*Do. sarmentorum*	CBS 128310	HQ288219	MH876299	HQ288263
*Do. sarmentorum*	CBS 141587	KX357188	N/A	KX357211
*Do. sarmentorum*	MFLUCC 17-0242	KY797637	KY815014	KY815020
*Do. sarmentorum*	CBS 115045	AY573202	AY928053	AY573222
*Do. sarmentorum*	CBS 113188	AY573198	EU673230	EU673278
*Do. sarmentorum*	MFLUCC 17-0951	MF398891	N/A	MF398943
*Do. sarmentorum*	CBS 140349	KP205497	N/A	KP205470
*Do. sarmentorum*	CBS 188.87	EU673316	DQ377891	EU673283
*Do. sarmentorum*	IRAN1579C	KC898234	N/A	KC898217
*Do. sarmentorum*	IRAN1585C	KC898235	N/A	KC898218
*Do. sarmentorum*	IRAN1583C	KC898236	N/A	KC898219
*Do. sarmentorum*	IRAN1581C	KC898237	N/A	KC898220
*Do. sarmentorum*	MFLUCC 13-0497	KJ742378	N/A	KJ742381
*Do. sarmentorum*	MFLUCC 13-0196	KU234782	N/A	KU234796
*Do. sarmentorum*	DAR78992	EU768874	N/A	EU768881
*Do. sarmentorum*	DAR78993	EU768876	N/A	EU768882
*Do. striata*	**ICMP 16824**	EU673320	EU673240	EU673287
*Do. striata*	ICMP 16819	EU673320	EU673239	EU673287
*Do. striata*	DAR 80992	KJ573643	N/A	KJ573640
*Do. styphnolobii*	**JZB3150013**	MH880849	N/A	MK069594
*Do. tectonae*	**MFLUCC 12-0382**	KM396899	N/A	KM409637
*Do. thailandica*	**MFLUCC 11-0438**	JX646796	JX646813	JX646861
*Do. thripsita*	**BRIP 51876**	FJ824738	N/A	KJ573639
*Do. ulmacea*	**CBS 138855**	KR611881	KR611899	KR611910
*Do. uruguayensis*	**CBS 124908**	EU080923	MH874932	EU863180
*Do. vinea gemmae*	**DAR81012**	KJ573644	N/A	KJ573641
*Do. viticola*	**CBS 117009**	AY905554	MH874565	AY905559
*Do. viticola*	CBS 117006	AY905555	EU673236	AY905562
*Do. viticola*	DAR80529	HM009376	N/A	HM800511
*Do. viticola*	DAR80530	HM009378	N/A	HM800513
*Do. yunnana*	**CGMCC 3.17999**	KX499643	N/A	KX499649
*Endomelanconiopsis endophytica*	**CBS 120397**	EU683656	EU683629	EU683637
*E. microspora*	**CBS 353.97**	EU683655	KF766330	EU683636
*Lasiodiplodia americana*	**CERC1960**	KP217058	N/A	KP217066
*L. americana*	CERC1961	KP217059	N/A	KP217067
*L. avicenniae*	**CMW41467**	KP860835	N/A	KP860680
*L. avicenniarum*	**MFLUCC 17-2591**	MK347777	MK347994	MK340867
*L. brasiliense*	**CMM4015**	JX464063	N/A	JX464049
*L. brasiliense*	CMM2185	KC484800	N/A	KC481530
*L. bruguierae*	**CMW42480**	KP860832	N/A	KP860677
*L. chonburiensis*	**MFLUCC 16-0376**		N/A	
*L. cinnamomi*	**CFCC 51997**	MG866028	N/A	MH236799
*L. citricola*	**CBS 124707**	GU945354	N/A	GU945340
*L. citricola*	IRAN1521C	GU945353	N/A	GU945339
*L. crassispora*	**CBS 118741**	DQ103550	DQ377901	EU673303
*L. crassispora*	WAC12534	DQ103551	N/A	DQ103558
*L. crassispora*	CBS 110492	EF622086	EU673251	EF622066
*L. crassispora*	CBS 121770	EU101307	N/A	EU101352
*L. crassispora*	CBS 121771	EU101308	N/A	EU101353
*L. euphorbicola*	**CMM3609**	KF234543	N/A	KF226689
*L. euphorbicola*	IBL329	KT247490	N/A	KT247492
*L. gilanensis*	**CBS 124704**	GU945351	N/A	GU945342
*L. gilanensis*	IRAN1501C	GU945352	N/A	GU945341
*L. gilanensis*	CBS 128311	HQ288225	N/A	HQ288267
*L. gilanensis*	UCD2199MO	HQ288226	N/A	HQ288268
*L. gonubiensis*	**CBS 115812**	AY639595	DQ377902	DQ103566
*L. gonubiensis*	CBS 116355	AY639594	EU673252	DQ103567
*L. gravistriata*	**CMM4564**	KT250949	N/A	KT250950
*L. hormozganensis*	**CBS 124709**	GU945355	N/A	GU945343
*L. hormozganensis*	CBS 124708	GU945356	N/A	GU945344
*L. iraniensis*	**CBS 124710**	GU945346	MH874918	GU945334
*L. iraniensis*	CBS 124711	GU945347	N/A	GU945335
*L. iraniensis*	CMM3610	MT587430	N/A	MT592142
*L. iraniensis*	CMM0247	MT587431	N/A	MT592143
*L. laeliocattleyae*	**CBS 167.28**	KU507487	N/A	KU507454
*L. laeliocattleyae*	CBS 130992	JN814397	N/A	JN814424
*L. laeliocattleyae*	CMM3611	JN814401	N/A	JN814428
*L. lignicola*	**CBS 134112**	JX646797	N/A	KU887003
*L. lignicola*	MFLUCC 11-0656	JX646797	JX646815	KU887003
*L. lignicola*	CGMCC 3.18061	KX499889	N/A	KX499927
*L. lignicola*	CBS 342.78	KX464140	N/A	KX464634
*L. macrospora*	**CMM3833**	KF234557	N/A	KF226718
*L. mahajangana*	**CBS 124927**	FJ900597	N/A	FJ900643
*L. mahajangana*	CBS 124925	FJ900595	N/A	FJ900641
*L. mahajangana*	IBL352	KT154759	N/A	KT154753
*L. mahajangana*	CMM1325	KT154760	N/A	KT008006
*L. mahajangana*	CBS 137785	KJ638317	N/A	KJ638336
*L. mahajangana*	BL184	KJ638318	N/A	KJ638337
*L. mahajangana*	MFLUCC 16-0265	MH275068	MH260301	MH412774
*L. margaritacea*	**CBS 122519**	EU144050	N/A	EU144065
*L. margaritacea*	CBS 122065	EU144051	N/A	EU144066
*L. marypalme*	**CMM2275**	KC484843	N/A	KC481567
*L. marypalme*	CMM2271	KC484844	N/A	KC481568
*L. mediterranea*	**CBS 137783**	KJ638312	N/A	KJ638331
*L. mediterranea*	ALG36	KJ638314	N/A	KJ638333
*L. parva*	CBS 456.78	EF622083	KF766362	EF622063
*L. parva*	CBS 494.78	EF622084	EU673258	EF622064
*L. parva*	CBS 356.59	AY343482	N/A	EF445396
*L. plurivora*	**CBS 120832**	EF445362	KX464356	EF445395
*L. plurivora*	CBS 121103	AY343482	KX464357	EF445396
*L. pontae*	**CMM1277**	KT151794	N/A	KT151791
*L. pontae*	IBL14	KT151794	N/A	KT151791
*L. pseudotheobromae*	CBS 116459	EF622077	EU673256	EF622057
*L. pseudotheobromae*	**CBS 447.62**	EF622081	MH869806	EF622060
*L. rubropurpurea*	**CBS 118740**	DQ103553	DQ377903	EU673304
*L. rubropurpurea*	WAC12536	DQ103554	N/A	DQ103572
*L. subglobosa*	**CMM3872**	KF234558	N/A	KF226721
*L. subglobosa*	CMM4046	KF234560	N/A	KF226723
*L. thailandica*	**CPC22795**	KJ193637	N/A	KJ193681
*L. thailandica*	CGMCC 3.17975	KX499879	MG321677	KX499917
*L. thailandica*	MFLUCC 18-0244	MK347789	N/A	MK340870
*L. theobromae*	**CBS 164.96**	AY640255	EU673253	AY640258
*L. theobromae*	CBS 124.13	DQ458890	AY928054	DQ458875
*L. theobromae*	CBS 111530	EF622074	N/A	EF622054
*L. theobromae*	CAA006	DQ458891	EU673254	DQ458876
*L. theobromae*	CBS 164.96	AY640255	EU673253	AY640258
*L. venezuelensis*	**CBS 118739**	DQ103547	DQ377904	EU673305
*L. venezuelensis*	WAC12540	DQ103548	N/A	DQ103569
*L. viticola*	**CBS 128313**	HQ288227	KX098286	HQ288269
*L. viticola*	CBS 128315	HQ288228	N/A	HQ288270
*L. vitis*	**CBS 124060**	KX464148	N/A	MN938928
*Lecanosticta acicola*	**LNPV252**	JX901755	JX901844	JX901639
*Macrophomina phaseolina*	**CBS 227.33**	KF951627	DQ377906	KF952000
*Neodeightonia palmicola*	**MFLUCC 10-0822**	HQ199221	HQ199222	N/A
*N. phoenicum*	**CBS 122528**	EU673340	EU673261	EU673309
*N. subglobosa*	**CBS 448.91**	EU673337	DQ377866	EU673306
*Neofusicoccum arbuti*	**CBS 116131**	AY819720	DQ377915	KF531792
*N. arbuti*	UW13	AY819724	N/A	KF531791
*N. arbuti*	CBS 117453	AY693976	DQ377914	AY693977
*N. arbuti*	CMW13446	DQ306263	N/A	DQ306264
*N. australe*	**CMW6837**	AY339262	KF766367	AY339270
*N. australe*	CMW6853	AY339263	N/A	AY339271
*N. brasiliense*	**CMM1338**	JX513630	N/A	JX513610
*N. buxi*	**CBS 116.75**	KX464164	KX464406	KX464677
*N. cordaticola*	**CBS 123634**	EU821898	MH874849	EU821868
*N. cordaticola*	CBS 123635	EU821903	KX464410	EU821873
*N. cryptoaustrale*	**CMW23785**	FJ752742	N/A	FJ752713
*N. eucalypticola*	**CBS 115679**	AY615141	N/A	AY615133
*N. eucalypticola*	CBS 115766	AY615143	N/A	AY615135
*N. eucalyptorum*	**CBS 115791**	AF283686	N/A	AY236891
*N. eucalyptorum*	CMW10126	AF283687	N/A	AY236892
*N. grevilleae*	**CBS 129518**	JF951137	N/A	N/A
*N. hellenicum*	**CERC1947**	KP217053	N/A	KP217061
*N. hongkongensis*	**CERC 2973**	KX278052	MF410096	KX278157
*N. illicii*	**CGMCC 3.18311**	KY350150	N/A	KY817756
*N. kwambonambiense*	**EU821900**	EU821900	N/A	EU821870
*N. kwambonambiense*	CMW14140	EU821919	N/A	EU821889
*N. lumnitzerae*	**CMW41469**	KP860881	N/A	KP860724
*N. luteum*	**CBS 110299**	AY259091	AY928043	AY573217
*N. luteum*	CBS 110497	EU673311	N/A	EU673277
*N. luteum*	CMW41365	KP860859	N/A	KP860702
*N. macroclavatum*	**CBS 118223**	DQ093196	N/A	DQ093217
*N. macroclavatum*	WAC12446	DQ093197	N/A	DQ093218
*N. mangiferae*	**CBS 118532**	AY615185	DQ377921	DQ093221
*N. mangiferae*	CMW7797	AY615186	N/A	DQ093220
*N. mediterraneum*	**CBS 121718**	GU251176	MH874696	GU251308
*N. mediterraneum*	CBS 121558	GU799463	N/A	GU799462
*N. mediterraneum*	CBS 113083	KX464186	KX464465	KX464712
*N. mediterraneum*	CBS 113089	KX464199	N/A	KX464727
*N. microconidium*	**CERC 3497**	KX278053	MF410097	KX278158
*N. nonquaesitum*	**CBS 126655**	GU251163	MH875645	GU251295
*N. nonquaesitum*	PD301	GU251164	N/A	GU251296
*N. occulatum*	**CBS 128008**	EU301030	MH876179	EU339509
*N. occulatum*	MUCC286	EU736947	N/A	EU339511
*N. parvum*	**CMW9081**	AY236943	AY928045	AY236888
*N. parvum*	CBS 110301	AY259098	AY928046	AY573221
*N. parvum*	CBS 137504	KJ657702	N/A	KJ657715
*N. parvum*	ALG9	KJ657704	N/A	KJ657721
*N. parvum*	MFLUCC 15-0900	KY856755	N/A	KY856754
*N. pennatisporum*	**WAC13153**	EF591925	EF591942	EF591976
*N. pistaciae*	**CBS 595.76**	KX464163	KX464404	KX464676
*N. protearum*	**STE-U 4361**	AF196295	N/A	N/A
*N. protearum*	STE-U 1775	AF452539	N/A	N/A
*N. ribis*	**CMW7772**	AY236935	N/A	AY236877
*N. ribis*	CBS 121.26	AF241177	N/A	AY236879
*N. ribis*	CBS 124923	FJ900607	N/A	FJ900653
*N. ribis*	CBS 124924	FJ900608	N/A	FJ900654
*N. ribis*	CMW14058	EU821904	N/A	EU821874
*N. ribis*	CMW14060	EU821905	N/A	EU821875
*N. sinense*	**CGMCC 3.18315**	KY350148	N/A	KY817755
*N. sinoeucalypti*	**CERC 2005**	KX278061	MF410105	KX278166
*N. stellenboschiana*	**CBS 110864**	AY343407	KX464513	AY343348
*N. terminaliae*	**CBS 125264**	GQ471802	N/A	GQ471780
*N. ursorum*	**CBS 122811**	FJ752746	MH874765	FJ752709
*N. viticlavatum*	**CBS 112878**	AY343381	MH874474	AY343342
*N. viticlavatum*	CBS 112977	AY343380	KX464528	AY343341
*N. vitifusiforme*	**CBS 110887**	AY343383	MH874455	AY343343
*N. vitifusiforme*	CBS 110880	AY343382	KX464475	AY343344
*N. vitifusiforme*	CBS 120081	DQ923533	MN162190	KX464682
*N. vitifusiforme*	CBS 121112	EF445349	N/A	EF445391
*Oblongocollomyces variabile*	**CBS 121774**	NR136994	KX464536	EU101357
*O. variabile*	CMW25420	EU101313	N/A	EU101358
*O. variabile*	CMW25421	EU101314	N/A	EU101359
*Phaeobotryon mamane*	CPC 12264	EU673331	DQ377898	EU673297
*P. mamane*	**CBS 122980**	EU673332	EU673248	EU673298
*Sardiniella celtidis*	**MFLUCC 17-981**	MF443249	N/A	MF443248
*Sa. urbana*	BL180	KX379677	KX379679	KX379678
*Sa. urbana*	BL181	KX379680	KX379682	KX379681
*Sa. urbana*	**CBS 141580**	KX379674	KX379676	KX379675
*Sphaeropsis citrigena*	**ICMP16812**	EU673328	EU673246	EU673294
*Sp. citrigena*	ICMP16818	EU673329	EU673247	EU673295
*Sp. eucalypticola*	**MFLUCC 11-0579**	JX646802	JX646819	JX646867
*Sp. eucalypticola*	MFLUCC 11-0654	JX646803	JX646820	JX646868
*Sp. porosa*	**CBS 110496**	AY343379	DQ377894	AY343340
*Sp. porosa*	CBS 110574	AY343378	DQ377895	AY343339
*Sp. visci*	**CBS 100163**	EU673324	N/A	EU673292
*Sp. visci*	CBS 186.97	EU673325	DQ377868	EU673293
*Tiarosporella graminis-karoo*	**CBS 118718**	KF531828	DQ377939	KF531807
*Melanopsaceae*				
*Melanops* sp.	CBS 118.39	FJ824771	DQ377856	FJ824776
*M. tulasnei*	**CBS 116805**	FJ824769	KF766365	FJ824774
*M. tulasnei*	CBS 116806	FJ824770	FJ824765	KX464644
*Phyllostictaceae*				
*Phyllosticta citricarpa*	**CBS 102374**	FJ538313	DQ377877	FJ538376
*P. minima*	**CBS 111635**	KF766215	EU754194	KF766433
*P. parthenocissi*	**CBS 111645**	EU683672	DQ377876	EU683653
*P. podocarpi*	**CBS 111647**	KF766217	KF766383	KF766434
*Pseudofusicoccum adansoniae*	**CMW 26147**	KF766220	KF766386	N/A
*P. ardesiacum*	**CMW 26159**	EU144060	KF766387	N/A
*P. kimberleyensis*	**CMW 26156**	EU144057	KF766388	N/A
*Planistromellaceae*				
*Kellermania agaves*	**CPC 21713**	KF777164	KF777217	N/A
*K. confusa*	**CBS 131723**	KF766174	KF766344	KF766405
*K. micranthae*	**CBS 131724**	KF766179	NG042706	KF766410
*K. plurilocularis*	**CBS 131719**	KF766181	KF766351	KF766412
*K. yuccifoliorum*	**CBS 131726**	KF766185	KF766355	KF766416
*Umthunziomyces hagahagensis*	**CPC 29917**	KY173472	KY173561	N/A
*Saccharataceae*				
*Neoseptorioides eucalypti*	**CBS 140665**	KT950857	KT950871	KT950882
*Saccharata banksiae*	**CPC 27698**	KY173449	KY173539	KY173596
*S. daviesiae*	**CPC 29174**	KY173450	KY173540	N/A
*S. proteae*	**CBS 115206**	KF766226	DQ377882	GU349030
*Septorioides pini-thunbergi*	**CBS 473.91**	NR145234	KF251746	N/A
*S. strobi*	**CBS 141443**	KT884699	KT884685	KT884713

ALG: Personal culture collection of A. Berraf-Tebbal; ATCC: American Type Culture Collection, Virginia, USA; BL: Personal collection of B.T. Linaldeddu; BRIP: Culture collection, Queensland Department of Agriculture and Fisheries, Queensland, Australia; CAA: Personal culture collection of Artur Alves, Universidade de Aveiro, Portuga; CBS: CBS-KNAW Fungal Biodiversity Centre, Utrecht, The Netherlands; CERC: Culture collection of China Eucalypt Research Centre, Chinese Academy of Forestry, ZhanJiang, GuangDong, China; CFCC: China Forestry Culture Collection Center, Beijing, China; CGMCC: China General Microbiological Culture Collection Center; CMM: Culture Collection of Phytopathogenic Fungi “Prof. Maria Menezes”, Universidade Federal Rural de Pernambuco, Recife, Brazil; CMW: Tree Patholgy Co-operative Program, Forestry and Agricultural Biotechnology Institute, University of Pretoria, South Africa; CPC: Working collection of P.W. Crous, housed at CBS; DAR: Plant Pathology Herbarium, Orange Agricultural Institute, Forest Road, Orange. NSW 2800, Australia; GZCC: Guizhou Academy of Agricultural Sciences Culture Collection, GuiZhou, China; IBL: Personal culture collection of I.B.L. Coutinho; ICMP: International Collection of Microorganisms from Plants, Landcare Research, Aukland, New Zealand; IMI: International Mycological Institute, CBI-Bioscience, Egham, Bakeham Lane, UK; IRAN: Iranian Fungal Culture Collection, Iranian Research Institute of Plant Protection, Iran; KUMCC: Kunming University Culture Collection, Yunnan, China. MFLU: Mae Fah Luang University Herbarium Collection, Chiang Rai, Thailand; MFLUCC: Mae Fah Luang University Culture Collection, Chiang Rai, Thailand; MUCC: Murdoch University Culture Collection, Murdoch, Australia; PD: Culture collection, University of California, Davis, USA; STE-U: Culture collection of the Department of Plant Pathology, University of Stellenbosch, South Africa; UCD: University of California, Davis, Plant Pathology Department Culture Collection; UCROK: Culture collection, University of Riverside, California, USA; WAC: Department of Agriculture, Western Australia Plant Pathogen Collection, South Perth, Western Australia.

**Table 4 jof-07-00893-t004:** Alignment details and comparison of MP, ML, and BI analyses results of each phylogenetic tree constructed in this study.

Character			*Botryosphaeriales* (Figure 1)	*Botryosphaeria* (Figure 10)	*Diplodia* (Figure 11)	*Dothiorella* (Figure 12)	*Lasiodiplodia* (Figure 13)	*Neofusicoccum* (Figure 14)
Number of base pairs in each gene region (including the gaps after alignment)			ITS (680), LSU (803), *tef* (374)	ITS (565), *tef* (265)	ITS (551), *tef* (331)	ITS (504), *tef* (343)	ITS (490), *tef* (344)	ITS (549), *tef* (253)
Number of isolates obtained in this study			12	23	6	2	5	12
Number of taxa originated from GenBank			95	32	49	68	78	65
Outgroup taxon			*Lecanosticta acicula*	*Macrophomina phaseolina*	*Lasiodiplodia theobromae*	*Neofusicoccum parvum*	*Diplodia mutila*	*Dothiorella viticola*
MP	Total number of characters		1857	830	882	847	834	802
	Constant characters		954	690	688	578	592	565
	Variable/parsimony uninformative characters		204	75	52	77	54	107
	Parsimony informative characters		704	70	146	202	192	134
	Number of parsimonious trees obtained		77	5	10	10	10	10
	Tree length (TL)		4199	180	385	725	559	463
	(CI)		0.393	0.911	0.644	0.568	0.612	0.646
	(RI)		0.762	0.938	0.865	0.835	0.857	0.865
	(RC)		0.300	0.854	0.557	0.475	0.524	0.559
	(HI)		0.607	0.089	0.356	0.432	0.388	0.354
ML	Final likelihood value		−22,205.648157	−2130.358718	−3417.346418	−4843.109271	−4183.640855	−3665.812286
	Number of distinct alignment patterns		1096	212	297	390	293	310
	Percentage of undetermined characters or gaps		26.52%	7.31%	11.38%	21.84%	8.52%	9.63%
	Base frequencies	A	0.223450	0.206023	0.207760	0.210373	0.213076	0.201476
		C	0.259348	0.298959	0.297528	0.293067	0.286249	0.297204
		G	0.288244	0.259350	0.261525	0.252521	0.258157	0.273045
		T	0.228958	0.235668	0.233187	0.244039	0.242519	0.228274
	Substitution rates	AC	1.251462	0.503912	0.982007	1.364862	0.729024	1.047005
		AG	2.644446	1.747452	3.644983	2.268590	2.724108	4.839820
		AT	1.415897	1.056533	0.863841	1.240011	1.053880	1.265619
		CG	1.353737	0.612519	1.681164	1.176948	0.817807	0.870409
		CT	4.677551	3.387698	4.528302	4.552526	4.062263	8.746955
		GT	1.000000	1.000000	1.000000	1.000000	1.000000	1.000000
	Gamma distribution rate parameter (alpha)		0.286904	0.231276	0.174245	0.234867	0.223197	0.268848
BI (model of each gene region)		ITS	SYM+I+G	GTR+I	GTR+I+G	HKY+I+G	SYM+I+G	GTR+I+G
		*tef*	HKY+G	GTR+G	HKY+I+G	GTR+G	HKY+I+G	HKY+G
Tree base ID			28690	28685	28686	28687	28688	28689
Reviewer access URL			http://purl.org/phylo/treebase/phylows/study/TB2:S28690?x-access-code=fe183dad30514d2dfbbfb8087dbbe53a&format=html	http://purl.org/phylo/treebase/phylows/study/TB2:S28685?x-access-code=ac877963f1b00fed7de4920660ebc78f&format=html	http://purl.org/phylo/treebase/phylows/study/TB2:S28686?x-access-code=9df25bff3d5c7c2faeb0029110c6675d&format=html	http://purl.org/phylo/treebase/phylows/study/TB2:S28687?x-access-code=382c7e58f8f7e3b815b9ae27dbe6f639&format=html	http://purl.org/phylo/treebase/phylows/study/TB2:S28688?x-access-code=e58fc08c292227a8965dd978203b449a&format=html	http://purl.org/phylo/treebase/phylows/study/TB2:S28689?x-access-code=7152a6819a1dd4ef1e743d4db1e0240c&format=html

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
