# Peer review of "Occurrence and Morpho-Molecular Identification of *Botryosphaeriales* Species from Guizhou Province, China"

_jof, 2021, doi:10.3390/jof7110893_

Round 1

Reviewer 1 Report

The authors did a good work in the study of Botryosphaeriales strains isolated from decaying wood in several sites in Karst region of Guizhou province. The novelty of the strains is adequately supported following morphological and molecular identification and phylogenetic analysis. However, the manuscript has some critical issues concerning the overall evaluation of the authors‘ findings in discussion. Furthermore there are numerous problems throughout the manuscript with the choice of words as well as with the grammar and syntax. All highlighted words and sentences without comments in the manuscript represent errors in language.

Author Response

Dear Reviewer,

Thanks very much for your efforts on our manuscript, we have revised our paper following your suggestions, as well as providing the response to the comments. The revised version has been carefully checked by a native English speaker and improved if it is necessary.

Please kindly see the response in the attached files.

Sincerely yours,

Jian-Kui Liu

Response to reviewers

Occurrence and morpho-molecular identification of Botryosphaeriales species from Guizhou Province, China

Thank you for the review and suggestions for improving this manuscript. We appreciate the comments and suggestions and have considered necessary changes. A revised manuscript has been prepared. Responses to reviewer comments are given in green.

In the revised manuscript, responses to reviewer 1 are highlighted in yellow while the responses to reviewer 2 are highlighted in blue.

Reviewer 1

The authors did a good work in the study of Botryosphaeriales strains isolated from decaying wood in several sites in Karst region of Guizhou province. The novelty of the strains is adequately supported following morphological and molecular identification and phylogenetic analysis. However, the manuscript has some critical issues concerning the overall evaluation of the authors‘ findings in discussion. Furthermore there are numerous problems throughout the manuscript with the choice of words as well as with the grammar and syntax. All highlighted words and sentences without comments in the manuscript represent errors in language.

  1. The abstract must be rewritten.

The abstract must be clear and precise stating briefly what has been done and what has been found. Furthermore, it contains numerous grammar and syntax errors.

Dear reviewer, Thank you so much for this comment. The abstract was revised stating a brief introduction to the order, methodology and overall results.

  1. Zhang et al. (2021) report 33 genera

Thank you for your comment. This was corrected in the manuscript.

  1. Botryosphaeriaceae is a large family

Thank you for your comment. This was corrected in the manuscript.

  1. the meaning is not clear

Thank you for your comment. This was corrected in the manuscript.

  1. In this paragraph the work previously conducted in China on Botryosphaeriales along with results and the aim of the present study are presented. The aim of the study is poorly presented.

Thank you for your comment. The sentence was revised.

  1. This is results.

Thank you for your comment. That sentence was removed from the manuscript.

  1. move this sentence at the end of the paragraph

Thank you for your comment. The sentence was moved to the end of the paragraph.

  1. The novel species are highlighted not their type status!

The details of the type species of novel taxa are given in bold.

  1. The Culture Collection accession number for the Type strains is highlighted!

The culture collection accession number for the type strains are given in bold.

  1. why two names?

These were recently synonymized under Botryosphaeria dothidea.

  1. why supposedly identified?

Thank you for your comment. The sentence was revised.

  1. Discussion is very weak and not well organized to bring out the contribution of this study. It is a mixture of introductory material for each genus found and results with some sporadic comments. An overall evaluation of the diversity, taxonomy, ecology and distribution of the taxa (known and novel) studied during this investigation in relation to what is already known should be presented.

The introductory phrases of the discussion were avoided as much as possible. An overall evaluation of the diversity and distribution of each species isolated in this study was provided where necessary.

  1. only the new species is assigned to the genus in this study. The other had already been assigned.

Thank you for your comment. The sentence was revised.

  1. How the sexual morph is distinguished by phylogenetic analysis?

Thank you for your comment. The sentence was revised.

  1. there are no data presented in the manuscript concerning the aetiology and epidemiology.   the authors studied Botryosphaeriales on decaying wood and there is no reference in the manuscript concerning any association of the isolated taxa to infected plants studied.  The meaning of this sentence is not evident. What do the authors mean by saying "raise queries about the source, overview and trail of these fungi"?

They suggest to take actions to discontinue their further spread. In the manuscript there are no data presented about the distribution of the species described and any association with widespread infections in the area studied.

The last paragraph of the discussion was revised according to these comments.

Reviewer 2 Report

Dear authors,

I have carefully revised the manuscript entitled “Occurrence and morpho-molecular identification of Botryosphaeriales species from Guizhou Province, China”, submitted for publication in Journal of Fungi, MPDI. This manuscript describes the identification of 60 isolates from the order Botryosphaeriales isolated from decayed woody samples and collected from several providence of China. Three of these 60 isolates are described for the first time, and others were reported for the first time in China. The identification of the isolated was assisted with comprehensive morfo-cultural features and by phylogenetic analysis.

Overall, the manuscript is well written and straightforward. The manuscript is divided into several sections (Introduction, Material and Methods, Results, Discussion, Conclusion and References), and results are well supported by figures, tables, taxonomic descriptions, and references.

However, there was one point that was not clear for me concerning selecting 2 or 3 genes for the phylogenetic analysis of the genera and order, respectively. This is more clearly explained in the following list to suggestions/corrections that I think can improve the manuscript.

Best regards

List to suggestions/corrections:

Note: The numbering of pages (and lines) starts only on page 6 of the manuscript. Therefore, I will refer to the section and paragraph for suggestions/corrections.

Section 1. Introduction:

1st paragraph –

  • “Many are also known to exist as endophytes in healthy plant tissues and as saprobes in dead tree materials.” – please rephrase to “Many are known to exist as endophytes in healthy plant tissues and also as saprobes in dead tree materials.”
  • “Hence, we follow this treatment in our study “ – please rephrase to “Hence we adopted/followed this last taxonomical revision. “

Section 2. Material and methods:

  • In section 2.1. the authors indicate that the isolates were obtained from decomposing woody hosts. Since in the 3rd paragraph in section 1. Introduction indicates that “(…) Botryosphaeriales contain species identified to cause blight, canker, dieback and fruit rots on a variety of woody perennials globally. (…)” could the authors include some extra information about the symptoms/disease, from each of the isolate, was obtained?
  • Please include the brand of the PDA used since PDA’s from different brands can produce different morphological features like colony colour. Include this information also in figure captions.
  • Section 2.2 – “(…)The BLAST tool (https://blast.ncbi.nlm.nih.gov/Blast.cgi) was incorporated to compare the attained (…) - please rephrase to “(…)The BLAST tool (https://blast.ncbi.nlm.nih.gov/Blast.cgi) was used to compare the obtained(…)
  • Section 2.2 – “(…)two additional gene regions, those encrypting translation elongation factor 1-a (tef) and large subunit rRNA gene (LSU) were sequenced. (...) – please rephrase to “(…)two additional gene regions, coding for translation elongation factor 1-a (tef) and large subunit rRNA gene (LSU) were sequenced. (…)”. Also, include a reference to justify the choice for these genes.
  • There is confusion about Tables 1 and 2 – in the text, table 1 concerns PCR conditions, and 2 concerns the isolates used in work.
  • Concerning the positive amplicons, amplification was single or multiband? If single, was the PCR product purified before sequencing? If multiband, the relevant one was extracted and how?
  • Section 2.3 – “(…) Sequences were attained by both forward and reverse primers (…) -please rephrase to “(…) Sequences were obtained by both forward and reverse primers (…)
  • Section 2.3 – end of the 1st paragraph – table 1, should be table 2? Please correct this.
  • Section 2.3 – it is not clear why the authors used only two genes (ITS and tef) for the phylogenic analysis of the genera and three genes for the phylogenetic analysis of the order, since for some reference type retrieved from the NCBI database and used to identify the isolates, there are 2/3 genes. Please better explain this choice.
  • Section 2.3 – table 1 (concerning the isolated used in this study) – include the meaning of N/A in the caption. Same for table 3.
  • Section 2.3 - Table 2 – please include the amplicon length since this information is helpful to others to identify the positive amplification.

Section 3. Results

Concerning this section, I don’t agree with the order of the figures. The “results” start with the description of the species/genera identified, but the figures don’t follow the same order. The first figure (figure 1) concerns the phylogenetic analysis of the order Botryosphaerialles, but this result follows the species/genera identified. It does not make sense and so please, correct this. It should be the species/genera first, followed by order analysis, in the text and the figures.

#line 32 – please rephrase to “All the details”

#line 37 – The remaining twelve isolates were treated separately because the blast was, probably, not conclusive. Therefore, the phylogenetic analysis of the order would be a way to identify these isolates.  If this is correct, I think that is now well explained. Please include a simple and clear explanation for this decision.

#line 48/49 – the phrase is misleading. It should be “In this phylogeny, five isolates clustered with Apolosporella hesperidica (1), Barriopsis tectonae (1), Sphaeropsis citrigena (1), Sardiniella celtidis (1) and Sa. Guizhouensis (1) respectively.” – Note. In figure 1, the Sardinella guizouensis cluster has 5 isolates. One, the GZCC 19-0229, is indicated in table 1, but for the others, there is no reference in the text or Tables 1 and 3. Is this correct?

Section Discussion:

#line 357 – “tweleve species” correct to “twelve species”

Author Response

Response to reviewers

Occurrence and morpho-molecular identification of Botryosphaeriales species from Guizhou Province, China

Thank you for the review and suggestions for improving this manuscript. We appreciate the comments and suggestions and have considered necessary changes. A revised manuscript has been prepared. Responses to reviewer comments are given in green.

In the revised manuscript, responses to reviewer 1 are highlighted in yellow while the responses to reviewer 2 are highlighted in blue.

Reviewer 2

I have carefully revised the manuscript entitled “Occurrence and morpho-molecular identification of Botryosphaeriales species from Guizhou Province, China”, submitted for publication in Journal of Fungi, MPDI. This manuscript describes the identification of 60 isolates from the order Botryosphaeriales isolated from decayed woody samples and collected from several providence of China. Three of these 60 isolates are described for the first time, and others were reported for the first time in China. The identification of the isolated was assisted with comprehensive morfo-cultural features and by phylogenetic analysis.

Overall, the manuscript is well written and straightforward. The manuscript is divided into several sections (Introduction, Material and Methods, Results, Discussion, Conclusion and References), and results are well supported by figures, tables, taxonomic descriptions, and references.

However, there was one point that was not clear for me concerning selecting 2 or 3 genes for the phylogenetic analysis of the genera and order, respectively. This is more clearly explained in the following list to suggestions/corrections that I think can improve the manuscript.

Dear Reviewer, thank you so much for your comments. An overview phylogenetic tree for the order Botryosphaeriales was constructed using ITS, tef and LSU. The reason to choose three genes is that some species in other families in Botryosphaeriales (except Botryosphaeriaceae) comprise only ITS and LSU sequences. As most of those species lack tef sequence data we followed Dissanayake et al. (2016) – ‘Botryosphaeriaceae: Current status of genera and species’ when constructing the order tree. Separate phylogenetic trees of the larger genera (Botryosphaeria, Diplodia, Dothiorella, Lasiodiplodia and Neofusicoccum) were constructed by combining ITS and tef1-α sequences.

List to suggestions/corrections:

Note: The numbering of pages (and lines) starts only on page 6 of the manuscript. Therefore, I will refer to the section and paragraph for suggestions/corrections.

Section 1. Introduction:

1st paragraph –

“Many are also known to exist as endophytes in healthy plant tissues and as saprobes in dead tree materials.” – please rephrase to “Many are known to exist as endophytes in healthy plant tissues and also as saprobes in dead tree materials.”
“Hence, we follow this treatment in our study “ – please rephrase to “Hence we adopted/followed this last taxonomical revision. “

Thank you so much for your comments. The sentences were rephrased accordingly.

Section 2. Material and methods:

In section 2.1. the authors indicate that the isolates were obtained from decomposing woody hosts. Since in the 3rd paragraph in section 1. Introduction indicates that “(…) Botryosphaeriales contain species identified to cause blight, canker, dieback and fruit rots on a variety of woody perennials globally. (…)” could the authors include some extra information about the symptoms/disease, from each of the isolate, was obtained?

Thank you for this comment. As the isolates were obtained from decaying woody hosts, any disease symptoms could not be observed.

Please include the brand of the PDA used since PDA’s from different brands can produce different morphological features like colony colour. Include this information also in figure captions.

The brand name of the PDA is ‘OXOID CM0139’. This was included in the manuscript.

Section 2.2 – “(…)The BLAST tool (https://blast.ncbi.nlm.nih.gov/Blast.cgi) was incorporated to compare the attained (…) - please rephrase to “(…)The BLAST tool (https://blast.ncbi.nlm.nih.gov/Blast.cgi) was used to compare the obtained(…)

Thank you so much for your comment. The sentence was rephrased accordingly.

Section 2.2 – “(…)two additional gene regions, those encrypting translation elongation factor 1-a (tef) and large subunit rRNA gene (LSU) were sequenced. (...) – please rephrase to “(…)two additional gene regions, coding for translation elongation factor 1-a (tef) and large subunit rRNA gene (LSU) were sequenced. (…)”. Also, include a reference to justify the choice for these genes.

Thank you so much for your comment. The sentence was rephrased accordingly, and a reference was added to manuscript to justify the choice of the gene regions.

There is confusion about Tables 1 and 2 – in the text, table 1 concerns PCR conditions, and 2 concerns the isolates used in work.

Thank you so much for your comment. This was corrected in the manuscript.

Concerning the positive amplicons, amplification was single or multiband? If single, was the PCR product purified before sequencing? If multiband, the relevant one was extracted and how?
Amplification was resulted in a single band. Hence, the PCR product was not purified.

Section 2.3 – “(…) Sequences were attained by both forward and reverse primers (…) -please rephrase to “(…) Sequences were obtained by both forward and reverse primers (…)
Thank you so much for your comment. The sentence was rephrased accordingly.

Section 2.3 – end of the 1st paragraph – table 1, should be table 2? Please correct this.
Thank you so much for your comment. This was corrected in the manuscript.

Section 2.3 – it is not clear why the authors used only two genes (ITS and tef) for the phylogenic analysis of the genera and three genes for the phylogenetic analysis of the order, since for some reference type retrieved from the NCBI database and used to identify the isolates, there are 2/3 genes. Please better explain this choice.

An overview phylogenetic tree for the order Botryosphaeriales was constructed using ITS, tef and LSU. The reason to choose three genes is that some species in other families in Botryosphaeriales (except Botryosphaeriaceae) comprise only ITS and LSU sequences. As most of those species lack tef sequence data we followed Dissanayake et al. (2016) – ‘Botryosphaeriaceae: Current status of genera and species’ when constructing the order tree. Separate phylogenetic trees of the larger genera (Botryosphaeria, Diplodia, Dothiorella, Lasiodiplodia and Neofusicoccum) were constructed by combining ITS and tef1-α sequences.

Section 2.3 – table 1 (concerning the isolated used in this study) – include the meaning of N/A in the caption. Same for table 3.

Thank you for the comment. The following was included to the table caption. N/A: No sequence available.

Section 2.3 - Table 2 – please include the amplicon length since this information is helpful to others to identify the positive amplification.

Thank you for the comment. This has been included in Table 4.

Section 3. Results

Concerning this section, I don’t agree with the order of the figures. The “results” start with the description of the species/genera identified, but the figures don’t follow the same order. The first figure (figure 1) concerns the phylogenetic analysis of the order Botryosphaerialles, but this result follows the species/genera identified. It does not make sense and so please, correct this. It should be the species/genera first, followed by order analysis, in the text and the figures.

This mistake was corrected in the manuscript. As the newly introduced species are residing in the order tree, we placed it as the Figure 1 and the details of the new species and rarely observed species were provided accordingly.

The other genera trees were placed after all figures (mentioned above) as those trees represent very common species isolated in this study.

#line 32 – please rephrase to “All the details”

This was revised in the manuscript.

#line 37 – The remaining twelve isolates were treated separately because the blast was, probably, not conclusive. Therefore, the phylogenetic analysis of the order would be a way to identify these isolates. If this is correct, I think that is now well explained. Please include a simple and clear explanation for this decision.

Yes. An explanation for this matter has been provided in above comments.

#line 48/49 – the phrase is misleading. It should be “In this phylogeny, five isolates clustered with Apolosporella hesperidica (1), Barriopsis tectonae (1), Sphaeropsis citrigena (1), Sardiniella celtidis (1) and Sa. Guizhouensis (1) respectively.” –

This was revised in the manuscript.

Note. In figure 1, the Sardinella guizouensis cluster has 5 isolates. One, the GZCC 19-0229, is indicated in table 1, but for the others, there is no reference in the text or Tables 1 and 3. Is this correct?

Yes. Only one isolate of Sa. guizhouensis was obtained in this study and others were included in Chen et al. (2021) ‘Additions to Karst Fungi 5: Sardiniella guizhouensis sp. nov. (Botryosphaeriaceae) associated with woody hosts in Guizhou province, China’ manuscript.

Section Discussion:

#line 357 – “tweleve species” correct to “twelve species”

This was revised in the manuscript.

Round 2

Reviewer 1 Report

The authors revised the manuscript according to the comments. The manuscript is substantially improved.

In Table 3, for those strains that two names are presented, it should be corrected and only the valid, current name for each strain should be written in order to avoid any confusion.

Author Response

Thank you so much for your review and suggestions to improve our manuscript.

The following changes were made to the recent revision.

According to the reviewer’s suggestion, those strains that two names are presented in Table 3 were corrected. The accepted valid current name for each strain was provided.

Reviewer 2 Report

Dear authors, 

You have addressed all my questions, and I'm satisfied with the answers provided. However, I still can't entirely agree with the figures' order, and therefore I leave the final decision to the editor. 

Best regards

Author Response

Thank you so much for your review and suggestions to improve our manuscript.

We did not rearrange the figure order as the current figure order reflects the importance of the major findings of the study. Hence the Fig. 10 to Fig. 14 were kept at last in the results section.

Thank you very much again for your efforts on our manuscript.